# *Homo naledi*, a new species of the genus *Homo* from the Dinaledi Chamber, South Africa

Lee R Berger[1,2]*, John Hawks[1,3], Darryl J de Ruiter[1,4], Steven E Churchill[1,5], Peter Schmid[1,6], Lucas K Delezene[1,7], Tracy L Kivell[1,8,9], Heather M Garvin[1,10], Scott A Williams[1,11,12], Jeremy M DeSilva[1,13], Matthew M Skinner[1,8,9], Charles M Musiba[1,14], Noel Cameron[1,15], Trenton W Holliday[1,16], William Harcourt-Smith[1,17,18], Rebecca R Ackermann[19], Markus Bastir[1,20], Barry Bogin[1,15], Debra Bolter[1,21], Juliet Brophy[1,22], Zachary D Cofran[1,23], Kimberly A Congdon[1,24], Andrew S Deane[1,25], Mana Dembo[1,26], Michelle Drapeau[27], Marina C Elliott[1,26], Elen M Feuerriegel[1,28], Daniel Garcia-Martinez[1,20,29], David J Green[1,30], Alia Gurtov[1,3], Joel D Irish[1,31], Ashley Kruger[1], Myra F Laird[1,11,12], Damiano Marchi[1,32], Marc R Meyer[1,33], Shahed Nalla[1,34], Enquye W Negash[1,35], Caley M Orr[1,36], Davorka Radovcic[1,37], Lauren Schroeder[1,19], Jill E Scott[1,38], Zachary Throckmorton[1,39], Matthew W Tocheri[40,41], Caroline VanSickle[1,3,42], Christopher S Walker[1,5], Pianpian Wei[1,43], Bernhard Zipfel[1]

[1]Evolutionary Studies Institute and Centre of Excellence in PalaeoSciences, University of the Witwatersrand, Johannesburg, South Africa; [2]School of Geosciences, University of the Witwatersrand, Johannesburg, South Africa; [3]Department of Anthropology, University of Wisconsin-Madison, Madison, United States; [4]Department of Anthropology, Texas A&M University, College Station, United States; [5]Department of Evolutionary Anthropology, Duke University, Durham, United States; [6]Anthropological Institute and Museum, University of Zurich, Zurich, Switzerland; [7]Department of Anthropology, University of Arkansas, Fayetteville, United States; [8]School of Anthropology and Conservation, University of Kent, Canterbury, United Kingdom; [9]Department of Human Evolution, Max Planck Institute for Evolutionary Anthropology, Leipzig, Germany; [10]Department of Anthropology/Archaeology and Department of Applied Forensic Sciences, Mercyhurst University, Erie, United States; [11]Center for the Study of Human Origins, Department of Anthropology, New York University, New York, United States; [12]New York Consortium in Evolutionary Primatology, New York, United States; [13]Department of Anthropology, Dartmouth College, Hanover, United States; [14]Department of Anthropology, University of Colorado Denver, Denver, United States; [15]School of Sport, Exercise and Health Sciences, Loughborough University, Loughborough, United Kingdom; [16]Department of Anthropology, Tulane University, New Orleans, United States; [17]Department of Anthropology, Lehman College, Bronx, United States; [18]Division of Paleontology, American Museum of Natural History, New York, United States; [19]Department of Archaeology, University of Cape Town, Rondebosch, South Africa; [20]Paleoanthropology Group, Museo Nacional de Ciencias Naturales, Madrid, Spain; [21]Department of Anthropology, Modesto Junior College, Modesto, United States; [22]Department of Geography and Anthropology, Louisiana State University, Baton Rouge, United States; [23]School of Humanities and Social Sciences, Nazarbayev University, Astana, Kazakhstan; [24]Department of Pathology and Anatomical Sciences, University of

*For correspondence:
Lee.Berger@wits.ac.za

**Competing interests:** The authors declare that no competing interests exist.

**Reviewing editors:** Johannes Krause, University of Tübingen, Germany; Nicholas J Conard, University of Tübingen, Germany

Missouri, Columbia, United States; [25]Department of Anatomy and Neurobiology, University of Kentucky College of Medicine, Lexington, United States; [26]Human Evolutionary Studies Program and Department of Archaeology, Simon Fraser University, Burnaby, Canada; [27]Department d'Anthropologie, Université de Montréal, Montréal, Canada; [28]School of Archaeology and Anthropology, Australian National University, Canberra, Australia; [29]Faculty of Sciences, Biology Department, Universidad Autònoma de Madrid, Madrid, Spain; [30]Department of Anatomy, Midwestern University, Downers Grove, United States; [31]Research Centre in Evolutionary Anthropology and Palaeoecology, Liverpool John Moores University, Liverpool, United Kingdom; [32]Department of Biology, University of Pisa, Pisa, Italy; [33]Department of Anthropology, Chaffey College, Rancho Cucamonga, United States; [34]Department of Human Anatomy and Physiology, University of Johannesburg, Johannesburg, South Africa; [35]Center for the Advanced Study of Human Paleobiology, George Washington University, Washington, United States; [36]Department of Cell and Developmental Biology, University of Colorado School of Medicine, Aurora, United States; [37]Department of Geology and Paleontology, Croatian Natural History Museum, Zagreb, Croatia; [38]Department of Anthropology, University of Iowa, Iowa City, United States; [39]Department of Anatomy, DeBusk College of Osteopathic Medicine, Lincoln Memorial University, Harrogate, United States; [40]Human Origins Program, Department of Anthropology, National Museum of Natural History, Smithsonian Institution, Washington, United States; [41]Department of Anthropology, Lakehead University, Thunder Bay, Canada; [42]Department of Gender and Women's Studies, University of Wisconsin-Madison, Madison, United States; [43]Department of Paleoanthropology, Institute of Vertebrate Paleontology and Paleoanthropology, Beijing, China

**Abstract** *Homo naledi* is a previously-unknown species of extinct hominin discovered within the Dinaledi Chamber of the Rising Star cave system, Cradle of Humankind, South Africa. This species is characterized by body mass and stature similar to small-bodied human populations but a small endocranial volume similar to australopiths. Cranial morphology of *H. naledi* is unique, but most similar to early *Homo* species including *Homo erectus*, *Homo habilis* or *Homo rudolfensis*. While primitive, the dentition is generally small and simple in occlusal morphology. *H. naledi* has humanlike manipulatory adaptations of the hand and wrist. It also exhibits a humanlike foot and lower limb. These humanlike aspects are contrasted in the postcrania with a more primitive or australopith-like trunk, shoulder, pelvis and proximal femur. Representing at least 15 individuals with most skeletal elements repeated multiple times, this is the largest assemblage of a single species of hominins yet discovered in Africa.

## Introduction

Fossil hominins were first recognized in the Dinaledi Chamber in the Rising Star cave system in October 2013. During a relatively short excavation, our team recovered an extensive collection of 1550 hominin specimens, representing nearly every element of the skeleton multiple times (*Figure 1*), including many complete elements and morphologically informative fragments, some in articulation, as well as smaller fragments many of which could be refit into more complete elements. The collection is a morphologically homogeneous sample that can be attributed to no previously-known hominin species. Here we describe this new species, *Homo naledi*. We have not defined *H. naledi* narrowly based on a single jaw or skull because the entire body of material has informed our understanding of its biology.

**eLife digest** Modern humans, or *Homo sapiens*, are now the only living species in their genus. But as recently as 100,000 years ago, there were several other species that belonged to the genus *Homo*. Together with modern humans, these extinct human species, our immediate ancestors and their close relatives, are collectively referred to as 'hominins'.

Now Berger et al. report the recent discovery of an extinct species from the genus *Homo* that was unearthed from deep underground in what has been named the Dinaledi Chamber, in the Rising Star cave system in South Africa. The species was named *Homo naledi*; 'naledi' means 'star' in Sotho (also called Sesotho), which is one of the languages spoken in South Africa.

The unearthed fossils were from at least 15 individuals and include multiple examples of most of the bones in the skeleton. Based on this wide range of specimens from a single site, Berger et al. describe *Homo naledi* as being similar in size and weight to a small modern human, with human-like hands and feet. Furthermore, while the skull had several unique features, it had a small braincase that was most similar in size to other early hominin species that lived between four million and two million years ago. *Homo naledi*'s ribcage, shoulders and pelvis also more closely resembled those of earlier hominin species than those of modern humans.

The *Homo naledi* fossils are the largest collection of a single species of hominin that has been discovered in Africa so far and, in a related study, Dirks et al. describe the setting and context for these fossils. However, since the age of the fossils remains unclear, one of the next challenges will be to date the remains to provide more information about the early evolution of humans and their close relatives.

Order Primates LINNAEUS 1758
Suborder Anthropoidea MIVART 1864
Superfamily Hominoidea GRAY 1825
Family Hominidae GRAY 1825
Tribe Hominini GRAY 1825
Genus *Homo* LINNAEUS 1758
*Homo naledi* sp. nov. urn:lsid:zoobank.org:pub:00D1E81A-6E08-4A01-BD98-79A2CEAE2411

## Etymology

The word *naledi* means 'star' in the Sotho language and refers to the Dinaledi Chamber's location within the Rising Star cave system.

## Locality

The Dinaledi chamber is located approximately 30 meters underground, within the Rising Star cave system at about 26°1′13″ S; 27°42′43″ E. The system lies within the Malmani dolomites, approximately 800 meters southwest of the well-known site of Swartkrans in the Cradle of Humankind World Heritage Site, Gauteng Province, South Africa.

## Horizon and associations

The present sample of skeletal material from the Dinaledi Chamber was recovered during two field expeditions, in November 2013 and March 2014.

Six specimens from an ex situ context can be identified as bird bones, and few fragmentary rodent remains have been recovered within the excavation area. Neither of these faunal constituents can presently be associated with the hominin fossil collection (*Dirks et al., 2015*).

Aside from these limited faunal materials, the Dinaledi collection is entirely composed of hominin skeletal and dental remains. The collection so far comprises 1550 fossil hominin specimens, this number includes 1413 bone specimens and 137 isolated dental specimens; an additional 53 teeth are present in mandibular or maxillary bone specimens. Aside from the fragmentary rodent teeth, all dental crowns (n = 179) are hominin, recovered both from surface collection and excavation. Likewise, aside from the few bird elements, all morphologically informative bone specimens are clearly hominin. In all cases where elements are repeated in the sample, they are morphologically homogeneous, with

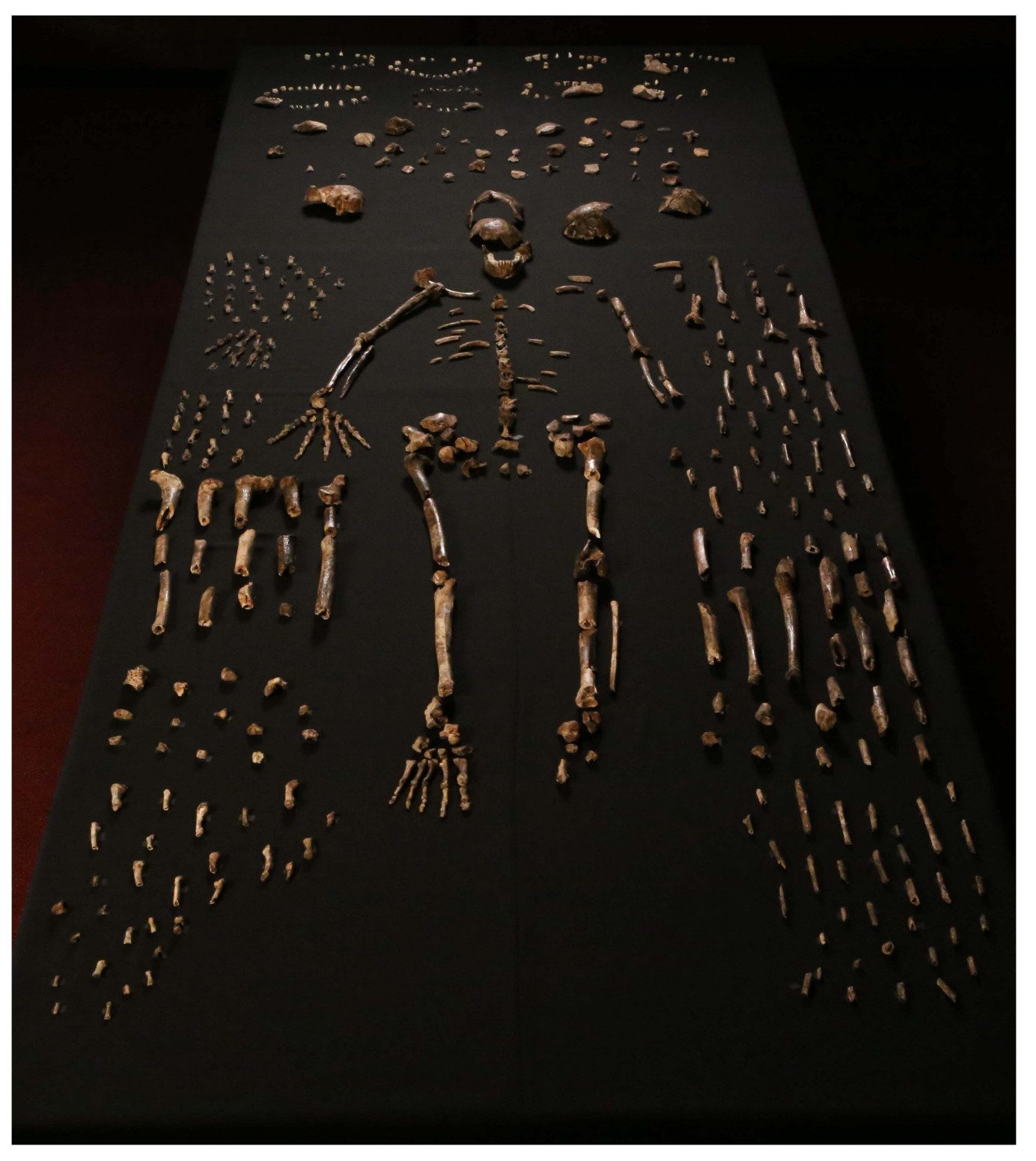

**Figure 1**. Dinaledi skeletal specimens. The figure includes approximately all of the material incorporated in this diagnosis, including the holotype specimen, paratypes and referred material. These make up 737 partial or complete anatomical elements, many of which consist of several refitted specimens. Specimens not identified to element, such as non-diagnostic long bone or cranial fragments, and a subset of fragile specimens are not shown here. The 'skeleton' layout in the center of the photo is a composite of elements that represent multiple individuals. This view is foreshortened; the table upon which the bones are arranged is 120-cm wide for scale.

variation consistent with body size and sex differences within a single population. These remains represent a minimum of 15 hominin individuals, as indicated by the repetition and presence of deciduous and adult dental elements.

The geological age of the fossils is not yet known. Excavations have thus far recovered hominin material from Unit 2 and Unit 3 in the chamber (*Dirks et al., 2015*). Surface-collected hominin material from the present top of Unit 3, which includes material derived from both Unit 2 and Unit 3, represents a minority of the assemblage, and is morphologically indistinguishable from material excavated from in situ within Unit 3. In addition to general morphological homogeneity including cranial shape, distinctive morphological configurations of all the recovered first metacarpals, femora, molars, lower premolars and lower canines, are identical in both surface-collected and excavated specimens (see Figure 14 later in the text). These include traits not found in any other hominin species yet described. These considerations strongly indicate that this material represents a single species, and not a commingled assemblage.

## Holotype, paratypes, and referred materials

### Holotype
Dinaledi Hominin 1 (DH1) comprises the partial calvaria, partial maxilla, and nearly complete mandible of a presumed male individual, based on size and morphology within the sample (*Figure 2*; *Supplementary file 1*). The holotype was recovered in situ during excavations within the Dinaledi Chamber in March of 2014, embedded in unconsolidated fine clay matrix (*Dirks et al., 2015*). The holotype is housed in the Evolutionary Studies Institute at the University of the Witwatersrand, Johannesburg, South Africa.

### Paratypes
Dinaledi Hominin 2 (DH2) is a partial calvaria that preserves parts of the frontal, left and right parietals, right temporal, and occipital (*Figure 3*; *Supplementary file 1*). Dinaledi Hominin 3 (DH3) is a partial

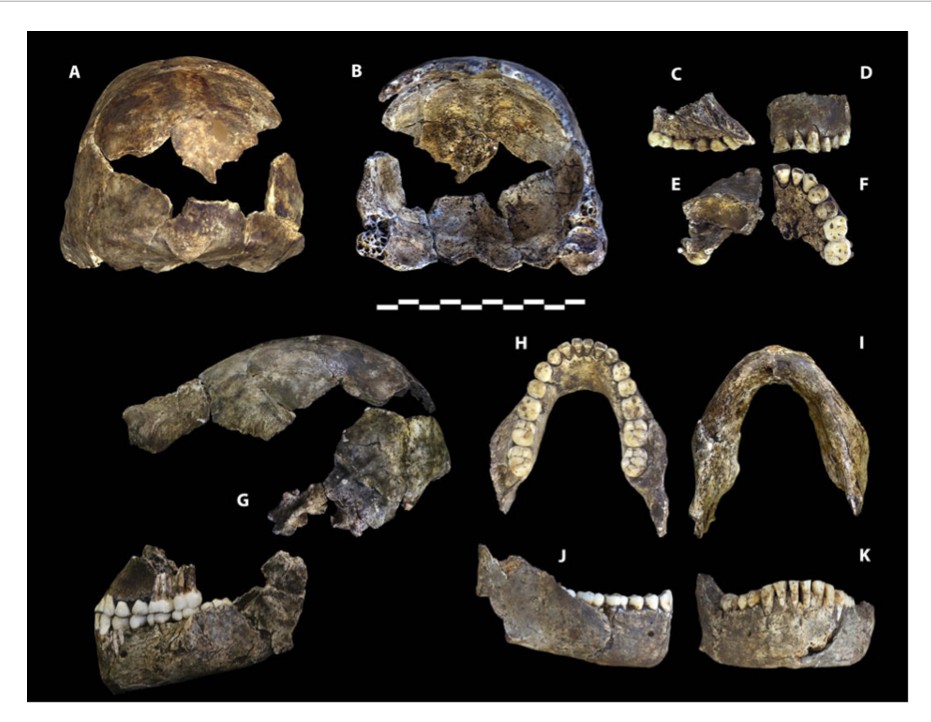

**Figure 2**. Holotype specimen of *Homo naledi*, Dinaledi Hominin 1 (DH1). U.W. 101-1473 cranium in (**A**) posterior and (**B**) frontal views (frontal view minus the frontal fragment to show calvaria interior). U.W. 101-1277 maxilla in (**C**) medial, (**D**) frontal, (**E**) superior, and (**F**) occlusal views. (**G**) U.W. 101-1473 cranium in anatomical alignment with occluded U.W. 101-1277 maxilla and U.W. 101-1261 mandible in left lateral view. U.W. 101-1277 mandible in (**H**) occlusal, (**I**) basal, (**J**) right lateral, and (**K**) anterior views. Scale bar = 10 cm.

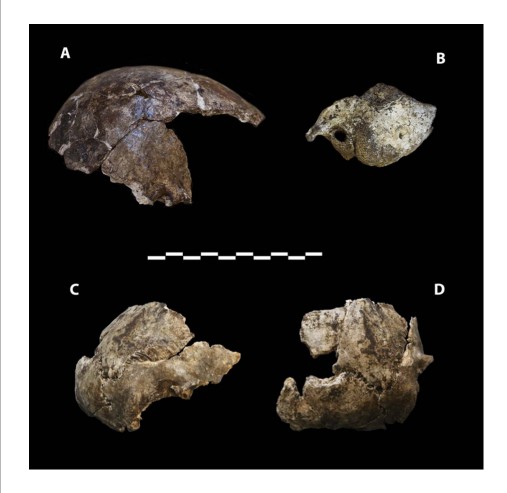

Figure 3. Cranial paratypes. (**A**) DH2, right lateral view. (**B**) DH5, left lateral view. (**C**) DH4, right lateral view. (**D**) DH4, posterior view. Scale bar = 10 cm.

calvaria of a presumed female individual that preserves parts of the frontal, left parietal, left temporal, and sphenoid (*Figure 4*, *Supplementary file 1*). Dinaledi Hominin 4 (DH4) is a partial calvaria that preserves parts of the right temporal, right parietal, and occipital (*Figure 3*; *Supplementary file 1*). Dinaledi Hominin 5 (DH5) is a partial calvaria that preserves part of the left temporal and occipital (*Figure 3*; *Supplementary file 1*). U.W. 101-377 is a mandibular fragment that preserves dental anatomy in an unworn state; at present it cannot be definitively associated with any of these Dinaledi Hominin (DH) individuals, and indeed might represent another individual (*Figure 5*; *Supplementary file 1*). These cranial specimens agree closely in nearly all morphological details where they overlap in areas preserved except those we interpret as related to sex.

Dinaledi hand 1 (H1) is a nearly complete (missing only the pisiform) right hand, found articulated in association, comprising specimens U.W. 101-1308 to −1311, −1318 to −1321, −1325 to −1329, −1351, −1464, and −1721 to −1732 (*Figure 6*; *Supplementary file 1*). U.W. 101-1391 is a proximal right femur preserving part of the head, the neck, some of the lesser and greater trochanter, and the proximal shaft (*Figure 7*; *Supplementary file 1*). U.W. 101-484 is a right tibial diaphysis missing only the proximal end (*Figure 8*; *Supplementary file 1*). Dinaledi foot 1 (F1) is a partial foot skeleton missing only the medial

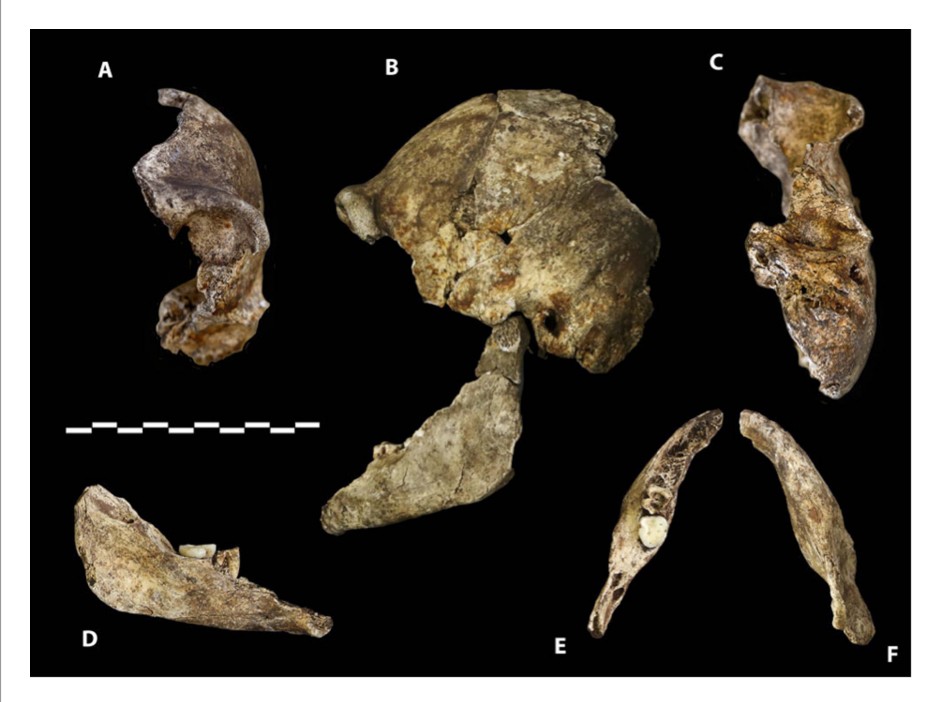

Figure 4. Paratype DH3. (**A**) Frontal view. (**B**) Left lateral view, with calvaria in articulation with the mandible (U.W. 101-361). (**C**) Basal view. Mandible in (**D**) medial view; (**E**) occlusal view; (**F**) basal view. DH3 was a relatively old individual at time of death, with extreme tooth wear. Scale bar = 10 cm.

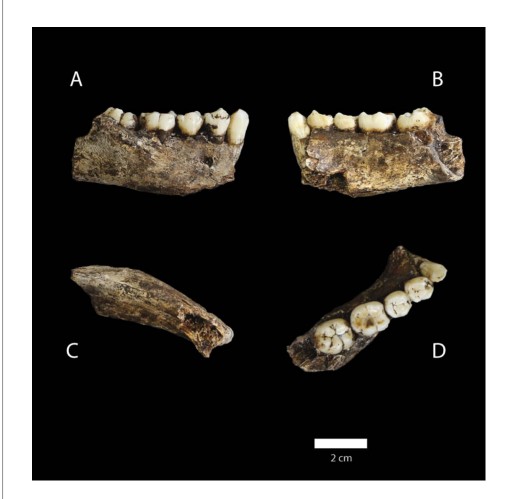

**Figure 5**. U.W. 101-377 mandible. (**A**) Lateral view; (**B**) medial view; (**C**) basal view; (**D**) occlusal view. (**D**) The distinctive mandibular premolar morphology with elongated talonids in unworn state. Scale bar = 2 cm.

cuneiform and the phalanges of rays II–V. Foot 1 is composed of specimens U.W. 101-1322, −1417 to −1419, −1439, −1443, −1456 to −1458, −1551, −1553, −1562, and −1698 (*Figure 9*; *Supplementary file 1*).

## Referred material
Referred material is also listed in *Supplementary file 1*. We refer to *H. naledi* all hominin material from the Dinaledi collection that can be identified to element; in total, the holotypes, paratypes and referred material comprise 737 partial or complete anatomical elements.

Specimen numbers in the collection are assigned at the point of excavation. Later laboratory analyses allowed us to refit specimens into more complete elements, which we have used as units of anatomical study. Here we refer to refitted elements by only a single specimen number; either the number of the most constitutive specimen, or the first diagnostic part to be discovered. DH designations are reserved for clearly associated individuals; at this time these are limited to the five partial crania designated above. Future excavation and analyses will certainly uncover more refits among specimens. As refits are found, all numbers assigned to refitted elements will remain stable, and all numbers in *Supplementary file 1* will be retained.

The collection is morphologically homogeneous in all duplicated elements, except for those anatomical features that normally reflect body size or sex differences in other primate taxa. Therefore, although we refer to the holotype and the paratypes for differential diagnoses; the section describing the overall anatomy encompasses all morphologically informative specimens.

## Differential diagnosis
This comprehensive differential diagnosis is based upon cranial, dental and postcranial characters. The hypodigms used for other species are detailed in the 'Materials and methods'. We examined original specimens for most species, except where indicated in the 'Materials and methods'; when we relied on other sources for anatomical observations we indicate this. A summary of traits of *H. naledi* in comparison to other species is presented in *Supplementary file 2*. Comparative cranial and mandibular measures are presented in *Table 1*, and comparative dental measures are provided in *Table 2*.

### Cranium, mandible, and dentition (DH1, DH2, DH3, DH4, DH5, U.W. 101-377)
The cranium of *H. naledi* does not have the well-developed crest patterns that characterize *Australopithecus garhi* (*Asfaw et al., 1999*) and species of the genus *Paranthropus*, nor the derived facial morphology seen in the latter genus. The mandible of *H. naledi* is notably more gracile than those of *Paranthropus*. Although maxillary and mandibular incisors and canines of *H. naledi* overlap in size with those of *Paranthropus*, the post-canine teeth are notably smaller than those of *Paranthropus* and *Au. garhi*, with mandibular molars that are buccolingually narrow.

*H. naledi* differs from *Australopithecus afarensis* and *Australopithecus africanus* in having a pentagonal-shaped cranial vault in posterior view, sagittal keeling, widely spaced temporal lines, an angular torus, a deep and narrow digastric fossa, an external occipital protuberance, an anteriorly positioned root of the zygomatic process of the maxilla, a broad palate, and a small canine jugum lacking anterior pillars. The anterior and lateral vault of *H. naledi* differs from *Au. afarensis* and *Au. africanus* in exhibiting only slight post-orbital constriction, frontal bossing, a well-developed supraorbital torus with a well-defined supratoral sulcus, temporal lines that are positioned on the

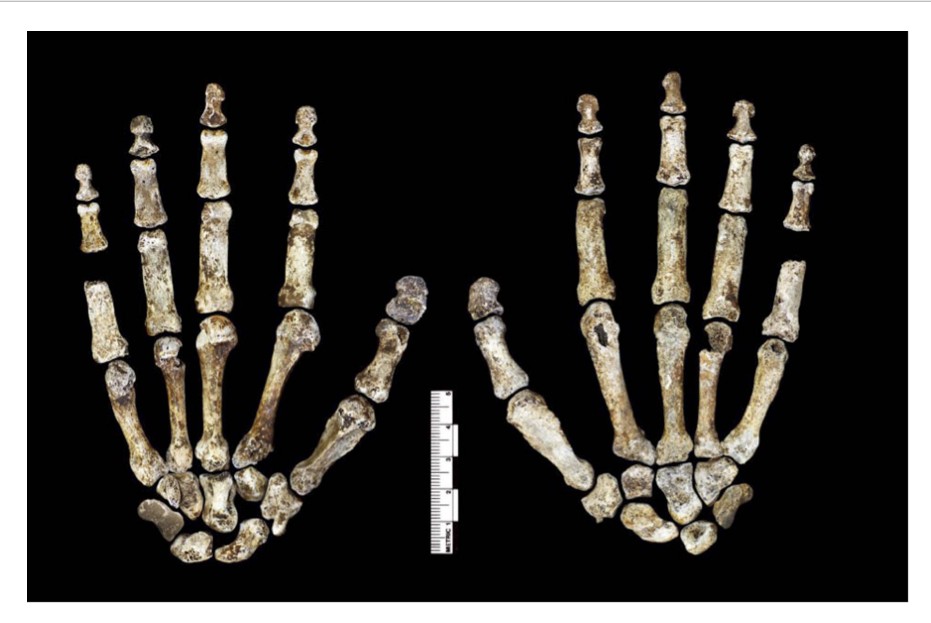

**Figure 6**. Hand 1. Palmar view on left; dorsal view on right. This hand was discovered in articulation and all bones are represented except for the pisiform. The proportions of digits are humanlike and visually apparent, as are the expanded distal apical tufts on all digits, the robust pollical ray, and the unique first metacarpal morphology.

posterior rather than the superior aspect of the supraorbital torus, a root of the zygomatic process of the temporal that is angled downwards approximately 30° relative to the Frankfort Horizontal (FH) and which begins its lateral expansion above the mandibular fossa rather than the EAM, a mandibular

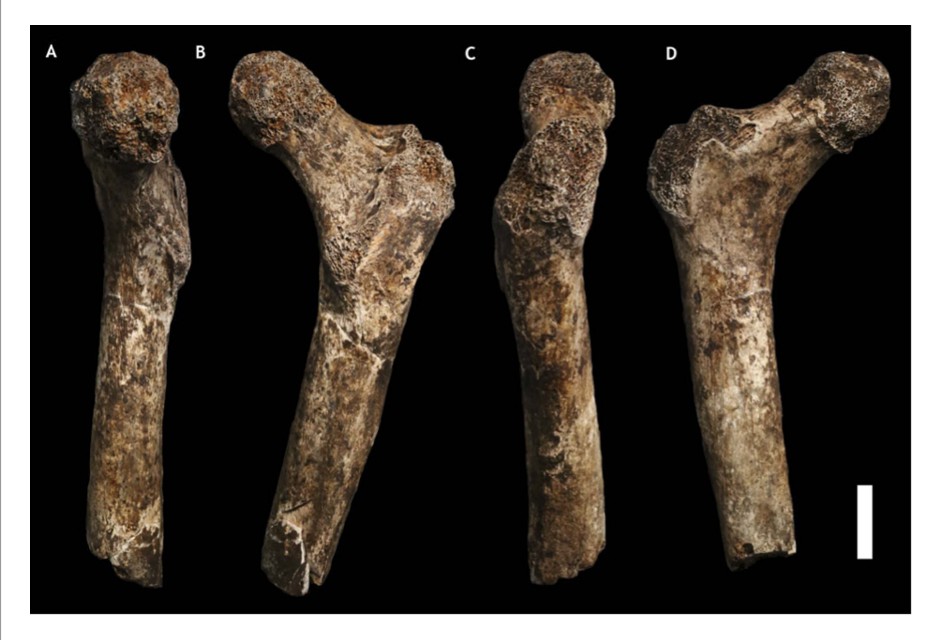

**Figure 7**. U.W. 101-1391 paratype femur. (**A**) Medial view; (**B**) posterior view; (**C**) lateral view; (**D**) anterior view. The femur neck is relatively long and anteroposteriorly compressed. The anteversion of the neck is evident in medial view. Scale bar = 2 cm.

fossa that is positioned medial to the wall of the temporal squame, a small postglenoid process that contacts the tympanic, a coronally oriented petrous, and a small and obliquely oriented EAM. The *H. naledi* mandible exhibits a more gracile symphysis and corpus, a more vertically inclined symphysis, a slight mandibular incurvation delineating a faint mental trigon, and a steeply inclined posterior face of the mandibular symphysis without a post incisive planum. The incisors of *H. naledi* overlap in size with some specimens of *Au. africanus*, though the canines and post-canine dentition are notably smaller, with relatively narrow buccolingual dimensions of the mandibular molars. The maxillary I[1] lacks a median lingual ridge and exhibits a broad and uninflated lingual cervical prominence, the lingual mesial and distal marginal ridges do not merge onto the cervical prominence in the maxillary I[2], the mandibular canine exhibits only a weak lingual median ridge and a broad and uninflated lingual cervical prominence, and the buccal grooves on the maxillary premolars are only weakly developed. *H. naledi* exhibits a small and isolated Carabelli's feature in the maxillary molars, unlike the more prominent and extensive Carabelli's feature of *Australopithecus*. Moreover, the *H. naledi* mandibular molars possess small, mesiobuccally restricted protostylids that do not intersect the buccal groove, differing from the typically enlarged, centrally positioned protostylids that intersect the buccal groove in *Australopithecus*.

The cranium of *H. naledi* differs from *Australopithecus sediba* (*Berger et al., 2010*) in exhibiting sagittal keeling, a more pronounced supraorbital torus and supratoral sulcus, a weakly arched supraorbital contour with rounded lateral corners, an angular torus, a well-defined supramastoid crest, a curved superior margin of the temporal squama, a root of the zygomatic process of the temporal that is angled downwards approximately 30° relative to FH, a flattened nasoalveolar clivus, weak canine juga, an anteriorly positioned root of the zygomatic process of the maxilla, and a relatively broad palate that is anteriorly shallow. The *H. naledi* mandible (DH1) has a mental foramen positioned superiorly on the corpus that opens posteriorly, unlike the mid-corpus height, more laterally opening mental foramen of *Au. sediba*. The maxillary and mandibular teeth of *H. naledi* are smaller than those of *Au. sediba*, with mandibular molars that are buccolingually narrow. The lingual mesial and distal marginal ridges do not merge onto the cervical prominence in the maxillary I[2], the paracone of the maxillary P[3] is equal in size to the protocone, the protoconid and metaconid of the mandibular molars are equally mesially positioned, and the lingual cusps of the molars are positioned at the occlusobuccal margin while the buccal cusps are positioned slightly lingual to the occlusobuccal margin. Also, *Au. sediba* shares with other australopiths a protostylid that is centrally located and which intersects the buccal groove of the lower molars, unlike the small and mesiobuccally restricted protostylid that does not intersect the buccal groove in *H. naledi*.

The cranium of *H. naledi* differs from *Homo habilis* in exhibiting sagittal keeling, a weakly arched supraorbital contour, temporal lines that are positioned on the posterior rather than the superior aspect of the supraorbital torus, an angular torus, an occipital torus, only slight postorbital constriction, a curved superior margin of the temporal squama, a suprameatal spine, a weak crista petrosa, a prominent Eustachian process, a small EAM, weak canine juga, and an anteriorly positioned root of the zygomatic process of the maxilla. Mandibles attributed to *H. habilis* show a weakly inclined, shelf-like post incisive planum with a variably developed superior transverse torus, unlike the steeply inclined posterior face of the mandibular symphysis of *H. naledi*, which lacks both a post incisive planum or

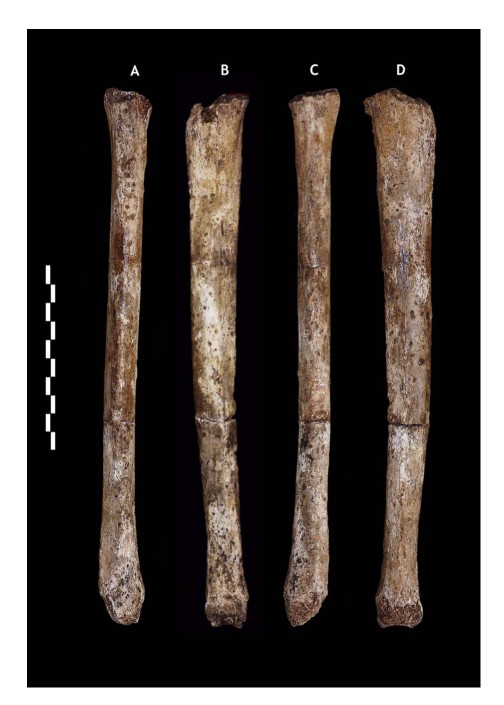

**Figure 8**. U.W. 101-484 paratype tibia. (**A**) Anterior view; (**B**) medial view; (**C**) posterior view; (**D**) lateral view. The tibiae are notably slender for their length. Scale bar = 10 cm.

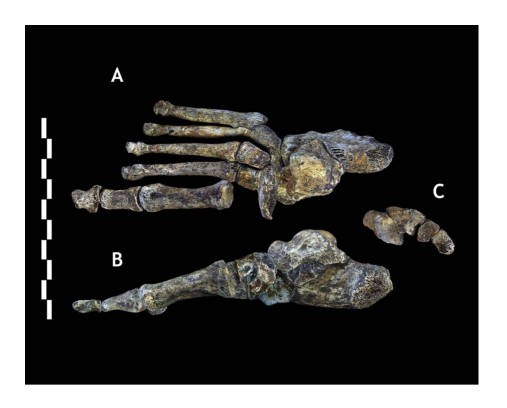

**Figure 9**. Foot 1 in (**A**) dorsal view; and (**B**) medial view. (**C**) Proximal articular surfaces of the metatarsals of Foot 1, shown in articulation to illustrate transverse arch structure. Scale bar = 10 cm.

superior transverse torus. The *H. naledi* mandible (DH1) has a mental foramen positioned superiorly on the corpus that opens posteriorly, while the mental foramen of *H. habilis* is at mid-corpus height and opens more laterally. The maxillary and mandibular dentitions of DH1 are smaller than typical for *H. habilis*. The mandibular $P_3$ of *H. naledi* is more molarized and lacks the occlusal simplification seen in *H. habilis*; it has a symmetrical occlusal outline, and multiple roots (two roots: mesiobuccal and distal) not seen in *H. habilis*. The molars of *H. naledi* lack crenulation, secondary fissures, and supernumerary cusps that are common to *H. habilis*. The protoconid and metaconid of the mandibular molars are equally mesially positioned.

The cranium of *H. naledi* differs from *Homo rudolfensis* by its smaller cranial capacity, and by exhibiting frontal bossing, a post-bregmatic depression, sagittal keeling, a well-developed supraorbital torus delineated by a distinct supratoral sulcus, temporal lines that are positioned on the posterior rather than the superior aspect of the supraorbital torus, an occipital torus, an external occipital protuberance, only slight post-orbital constriction, a small postglenoid process, a weak crista petrosa, a laterally inflated mastoid process, a canine fossa, incisors that project anteriorly beyond the bi-canine line, and a shallow anterior palate. As in *H. habilis*, mandibles attributed to *H. rudolfensis* show a weakly inclined, shelf-like post incisive planum with a variably developed superior transverse torus, unlike the steeply inclined posterior face of the mandibular symphysis of DH1, the latter of which lacks either a post incisive planum or superior transverse torus. The mandibular symphysis and corpus of *H. naledi* are more gracile than those attributed to *H. rudolfensis*, and the *H. naledi* mandible (DH1) has a mental foramen positioned superiorly on the corpus that opens posteriorly, unlike the mid-corpus height, more laterally opening mental foramen of *H. rudolfensis*. The maxillary and mandibular dentition of *H. naledi* is smaller than that of most specimens of *H. rudolfensis*, with only KNM-ER 60000 and KNM-ER 62000 appearing similar in size for some teeth (*Leakey et al., 2012*). The molars of *H. naledi* lack crenulation, secondary fissures, or supernumerary cusps common in *H. rudolfensis*. The buccal grooves of the maxillary premolars are weak in *H. naledi*, and the protoconid and metaconid of the mandibular molars are equally mesially positioned.

*H. naledi* lacks the typically distinctive long and low cranial vault of *Homo erectus*, as well as the metopic keeling that is typically present in the latter species. *H. naledi* also differs from *H. erectus* in having a distinct external occipital protuberance in addition to the tuberculum linearum, a laterally inflated mastoid process, a flat and squared nasoalveolar clivus, and an anteriorly shallow palate. The parasagittal keeling that is present between bregma and lambda in *H. naledi* (DH1 and DH3) is less marked than often occurs in *H. erectus*, including in small specimens such as KNM-ER 42700 and the Dmanisi cranial sample. Also unlike most specimens of *H. erectus*, *H. naledi* has a small vaginal process, a weak crista petrosa, a marked Eustachian process, and a small EAM. The mandible of *H. erectus* shows a moderately inclined, shelf-like post incisive planum terminating in a variably developed superior transverse torus, differing from the steeply inclined posterior face of the *H. naledi* mandibular symphysis, which lacks both a post incisive planum or a superior transverse torus. The mental foramen is positioned superiorly and opens posteriorly in DH1, unlike the mid-corpus height, more laterally opening mental foramen of *H. erectus*. The maxillary and mandibular incisors and canines of *H. naledi* are smaller than typical of *H. erectus*. The mandibular $P_3$ of *H. naledi* is more molarized and lacks the occlusal simplification seen in *H. erectus*, they reveal a symmetrical occlusal outline, and multiple roots (2R: MB+D) not typically seen in *H. erectus*. Furthermore, the molars of *H. naledi* lack crenulation, secondary fissures, or supernumerary cusps common in *H. erectus*.

*H. naledi* lacks the reduced cranial height of *Homo floresiensis*, and displays a marked angular torus and parasagittal keeling between bregma and lambda that is absent in the latter species. *H. naledi*

**Table 1.** Cranial and mandibular measurements for *H. naledi*, early hominins, and modern humans

| | Measurement definitions as in Wood (1991) | *P. aethiopicus* | *P. boisei* | *P. robustus* | *Au. afarensis* | *Au. africanus* | *Au. sediba* | *H. naledi* | *H. habilis* | *H. rudolfensis* | *H. erectus* | MP *Homo* | *H. sapiens* |
|---|---|---|---|---|---|---|---|---|---|---|---|---|---|
| **Cranium** | | | | | | | | | | | | | |
| Cranial capacity | – | 410 | 485 | 493 | 457 | 467 | 420 | 513 | 610 | 776 | 865 | 1266 | 1330 |
| Porion height | 6 | 72 | 74 | – | 86 | 70 | 67 | 81 | 77 | 90 | 94 | 101 | 112 |
| Posterior cranial length | 3 | 58 | 47 | 54 | 60 | 44 | – | 65 | 60 | 70 | 79 | 99 | 81 |
| Bi-parietal breadth | 9 | 94 | 98 | – | 90 | 99 | 100 | 103 | 107 | 118 | 129 | 142 | 132 |
| Bi-temporal breadth | 10 | 110 | 109 | 108 | 115 | 104 | 101 | 107 | 112 | 126 | 131 | 146 | 127 |
| Closest approach of temporal lines | – | crest* | crest* | crest* | crest* | 21 | 56 | 52 | 35 | 51 | 72 | 101 | 96 |
| Supraorbital height index | – | 46 | 53 | 50 | 51 | 60 | 56 | 56 | 64 | 59 | 56 | 62 | 71 |
| Minimum post-orbital breadth | – | 62 | 66 | 70 | 77 | 67 | 70 | 68 | 75 | 78 | 89 | 96 | 97 |
| Superior facial breadth | 49 | 100 | 107 | 109 | – | 95 | 86 | 86 | 97 | 113 | 110 | 124 | 107 |
| Post-orbital constriction index† | – | 62 | 61 | 64 | – | 69 | 81 | 79 | 72 | 74 | 81 | 80 | 91 |
| EAM area (as an ellipse)‡ | – | 77 | 80 | 103 | 70 | 96 | – | 38 | 76 | – | 95 | 85 | 61 |
| Root of zygomatic process origin | – | P4 | P4 | P3 to M1 | P4 to M1 | P4 to M1 | P4 | P3 to P4 | P4 to M1 | P4 to M1 | P4 to M1 | M1 | M1 |
| Petromedian angle | 137 | 50 | 45 | 50 | 31 | 33 | – | 55 | 48 | – | 52 | 55 | 46 |
| **Maxilloalveolar process** | | | | | | | | | | | | | |
| Maxilloalveolar length | 87 | 94 | 78 | 69 | 67 | 71 | 63 | 57 | 65 | 68 | 66 | 69 | 55 |
| Maxilloalveolar breadth | 88 | 83 | 76 | 69 | 68 | 66 | 63 | 71 | 68 | 72 | 70 | 72 | 62 |
| Palate breadth | 91 | 32 | 40 | 35 | 30 | 36 | 29 | 44 | 38 | 40 | 38 | 56 | 40 |

*Table 1. Continued on next page*

Table 1. Continued

| Measurement definitions as in Wood (1991) | | P. aethiopicus | P. boisei | P. robustus | Au. afarensis | Au. africanus | Au. sediba | H. naledi | H. habilis | H. rudolfensis | H. erectus | MP Homo | H. sapiens |
|---|---|---|---|---|---|---|---|---|---|---|---|---|---|
| Palate depth at incisive fossa | – | 3 | 11 | 10 | 10 | 9 | 10 | 5 | 10 | 13 | 11 | 10 | 9 |
| Palate depth at M1 | 103 | 7 | 18 | 11 | 11 | 13 | 10 | 10 | 12 | 16 | 15 | 18 | 13 |
| Mandible | | | | | | | | | | | | | |
| Symphysis height | 141 | 37 | 49 | 42 | 39 | 37 | 32 | 33 | 31 | 37 | 35 | 34 | 34 |
| Symphysis width | 142 | 26 | 28 | 25 | 20 | 21 | 18 | 18 | 20 | 24 | 18 | 17 | 14 |
| Symphysis area at M1 (as an ellipse)‡ | 146 | 757 | 1114 | 835 | 623 | 606 | 452 | 467 | 393 | 723 | 519 | 474 | 365 |
| Corpus height at M1 | 150 | 38 | 42 | 36 | 34 | 32 | 30 | 26 | 29 | 36 | 31 | 31 | 28 |
| Corpus breadth at M1 | 151 | 25 | 29 | 26 | 20 | 21 | 18 | 16 | 20 | 22 | 19 | 19 | 13 |
| Corpus area at M1 (as an ellipse)‡ | 152 | 742 | 955 | 736 | 540 | 539 | 405 | 326 | 425 | 631 | 458 | 469 | 296 |
| Mental foramen height index§ | – | 51 | 50 | 54 | 58 | 53 | 50 | 40 | 46 | 49 | 48 | 48 | 50 |

*At least in presumed males.

†Post-orbital breadth/superior facial breadth × 100.

‡Following the formula (π × (corpus height/2) × (corpus breadth/2)).

§Height of mental foramen from alveolar border relative to corpus height at the mental foramen.

MP, Middle Pleistocene.

Unless otherwise indicated measurements are defined as in **Wood (1991)**. Chord distances are in mm. Data for *H. naledi* collected from original fossils or laser scans by DJdeR and HMG; comparative data collected by DJdeR on original fossils and casts and supplemented by data from **Wood (1991)**.

**Table 2.** Dental measures for *H. naledi* and comparative hominin species

**Maxillary**

| | | I¹ | | I² | | C | | P³ | | P⁴ | | M¹ | | M² | | M³ | |
|---|---|---|---|---|---|---|---|---|---|---|---|---|---|---|---|---|---|
| | | MD | LL | MD | LL | MD | LL | MD | BL | MD | BL | MD | BL | MD | BL | MD | BL |
| *Au. anamensis* | n | 3 | 5 | – | 2 | 6 | 7 | 7 | 6 | 5 | 3 | 12 | 10 | 10 | 8 | 9 | 8 |
| | mean | 10.8 | 8.7 | – | 7.3 | 11.0 | 10.6 | 9.9 | 12.6 | 8.9 | 13.6 | 11.5 | 12.9 | 13.0 | 14.4 | 12.5 | 14.2 |
| | range | 9.1–12.4 | 8.2–9.3 | – | 7.0–7.5 | 9.9–12.3 | 9.1–11.8 | 8.2–11.8 | 10.1–14.3 | 7.2–12.1 | 12.6–14.2 | 7.8–14.3 | 9.0–16.7 | 10.9–16.3 | 12.9–16.1 | 11.1–15.7 | 13.0–15.7 |
| *Au. afarensis* | n | 7 | 8 | 9 | 9 | 15 | 15 | 12 | 10 | 18 | 12 | 16 | 13 | 10 | 11 | 11 | 11 |
| | mean | 10.7 | 8.4 | 7.5 | 7.2 | 9.9 | 10.8 | 8.8 | 12.4 | 9.1 | 12.4 | 12.0 | 13.4 | 12.9 | 14.6 | 12.7 | 14.5 |
| | range | 9.9–11.8 | 7.1–9.7 | 6.6–8.2 | 6.2–8.1 | 8.8–11.6 | 9.3–12.5 | 7.7–9.7 | 11.3–13.4 | 7.6–10.8 | 11.1–14.5 | 10.5–13.8 | 12.0–15.0 | 12.1–13.6 | 13.4–15.2 | 10.9–14.8 | 13.1–16.3 |
| *Au. africanus* | n | 15 | 15 | 11 | 10 | 16 | 13 | 26 | 25 | 20 | 20 | 21 | 20 | 23 | 24 | 27 | 28 |
| | mean | 10.7 | 8.3 | 6.9 | 6.8 | 9.9 | 10.3 | 9.2 | 12.7 | 9.5 | 13.4 | 12.9 | 13.9 | 14.1 | 15.7 | 14.2 | 16.0 |
| | range | 9.4–12.5 | 7.4–9.1 | 5.8–8.0 | 5.6–7.9 | 8.8–11.0 | 8.7–12.0 | 8.5–10.2 | 10.7–14.5 | 7.2–11.0 | 12.4–15.3 | 11.7–14.4 | 12.9–15.3 | 12.1–16.3 | 12.8–17.9 | 11.2–16.9 | 13.1–18.6 |
| *Au. sediba* | n | 1 | 1 | 1 | 1 | 1 | 1 | 1 | 1 | 1 | 1 | 1 | 1 | 1 | 1 | 2 | 2 |
| | mean | 10.1 | 6.9 | 7.2 | 6.6 | 9.0 | 8.8 | 9.0 | 11.2 | 9.3 | 12.1 | 12.9 | 12.0 | 12.9 | 13.7 | 13.0 | 13.5 |
| | range | – | – | – | – | – | – | – | – | – | – | – | – | – | – | 12.6–13.3 | 12.9–14.1 |
| *H. naledi* | n | 5 | 5 | 4 | 8 | 10 | 9 | 10 | 10 | 7 | 7 | 12 | 13 | 11 | 9 | 7 | 7 |
| | mean | 9.4 | 6.5 | 6.6 | 6.2 | 8.1 | 8.6 | 8.0 | 10.5 | 8.1 | 11.0 | 11.6 | 11.7 | 12.2 | 12.8 | 11.6 | 12.4 |
| | range | 8.8–9.8 | 6.3–7.0 | 6.3–7.0 | 5.8–6.6 | 7.3–8.9 | 8.0–9.6 | 7.7–8.4 | 9.8–11.0 | 7.7–8.7 | 10.5–12.4 | 10.5–12.4 | 11.2–12.4 | 11.0–13.0 | 11.9–13.6 | 11.0–12.7 | 11.4–13.4 |
| *H. habilis* | n | 2 | 2 | 4 | 4 | 2 | 3 | 7 | 7 | 8 | 8 | 13 | 13 | 7 | 7 | 7 | 7 |
| | mean | 10.6 | 8.0 | 7.4 | 6.6 | 9.0 | 9.8 | 9.0 | 11.9 | 9.2 | 12.1 | 12.7 | 13.0 | 12.7 | 14.3 | 12.3 | 14.7 |
| | range | 10.1–11.1 | 7.3–8.7 | 6.7–8.1 | 6.0–7.9 | 8.5–9.4 | 8.5–11.6 | 8.1–9.6 | 11.0–12.7 | 8.5–9.9 | 11.0–12.7 | 11.6–13.9 | 12.1–14.1 | 11.8–13.5 | 13.5–16.2 | 11.3–13.9 | 13.2–16.6 |
| *H. rudolfensis* | n | 1 | 1 | – | – | 1 | 1 | 1 | 1 | 2 | 2 | 2 | 2 | 2 | 2 | 1 | 1 |
| | mean | 12.3 | 7.7 | – | – | 11.5 | 12.5 | 10.5 | 13.6 | 10.2 | 12.5 | 14.0 | 14.0 | 14.3 | 15.8 | 13.3 | 13.5 |
| | range | – | – | – | – | – | – | – | – | 9.7–10.7 | 11.1–13.8 | 13.9–14.2 | 13.3–14.8 | 14.1–14.6 | 14.1–17.6 | – | – |
| *H. erectus* | n | 11 | 12 | 6 | 6 | 12 | 12 | 27 | 27 | 30 | 29 | 34 | 32 | 22 | 22 | 16 | 16 |
| | mean | 10.3 | 8.1 | 7.7 | 8.0 | 9.5 | 10.0 | 8.5 | 11.8 | 8.1 | 11.6 | 12.2 | 13.2 | 12.0 | 13.3 | 16 | 16 |
| | range | 8.1–12.6 | 7.0–11.7 | 6.0–8.3 | 6.9–8.5 | 8.5–11.1 | 9.0–11.1 | 7.1–10.1 | 9.5–13.8 | 7.0–9.4 | 9.9–13.4 | 10.1–14.6 | 11.0–15.9 | 10.3–13.6 | 10.9–15.5 | 8.7–14.7 | 10.4–15.8 |
| *H. neanderthalensis* | n | 28 | 37 | 35 | 41 | 28 | 29 | 16 | 17 | 21 | 19 | 23 | 24 | 27 | 28 | 22 | 21 |
| | mean | 9.7 | 8.5 | 8.0 | 8.4 | 8.8 | 10.1 | 8.0 | 10.6 | 7.8 | 10.6 | 11.6 | 12.3 | 10.9 | 12.5 | 9.9 | 12.3 |
| | range | 8.2–11.8 | 7.3–9.9 | 5.8–9.3 | 5.8–9.9 | 7.2–10.0 | 7.6–11.4 | 6.6–9.3 | 8.4–11.8 | 5.9–11.5 | 8.3–11.7 | 9.5–13.5 | 11.0–14.2 | 8.9–15.9 | 10.8–14.6 | 8.2–11.4 | 9.8–14.6 |

*Table 2. Continued on next page*

*Table 2. Continued*

## Maxillary

| | | I¹ | | I² | | C | | P³ | | P⁴ | | M¹ | | M² | | M³ | |
|---|---|---|---|---|---|---|---|---|---|---|---|---|---|---|---|---|---|
| | | MD | LL | MD | LL | MD | LL | MD | BL | MD | BL | MD | BL | MD | BL | MD | BL |
| *H. heidelbergensis* | n | 21 | 23 | 19 | 21 | 27 | 29 | 25 | 25 | 22 | 23 | 25 | 24 | 24 | 23 | 26 | 27 |
| | mean | 9.6 | 7.8 | 7.7 | 7.8 | 8.8 | 9.8 | 7.9 | 10.6 | 7.6 | 10.3 | 11.2 | 11.9 | 10.2 | 12.3 | 8.9 | 11.6 |
| | range | 8.7–10.7 | 7.1–9.9 | 7.2–8.4 | 7.3–8.6 | 8.1–11.0 | 8.8–11.8 | 7.1–9.0 | 9.2–12.2 | 7.0–8.8 | 9.1–11.5 | 9.9–12.3 | 10.3–13.2 | 8.1–12.1 | 11.1–13.8 | 7.6–11.0 | 10.0–13.2 |
| MP/LP African Homo | n | 6 | 6 | 7 | 8 | 4 | 4 | 6 | 6 | 10 | 10 | 14 | 14 | 20 | 20 | 9 | 9 |
| | mean | 9.0 | 7.8 | 7.4 | 7.2 | 8.9 | 9.7 | 8.4 | 10.8 | 8.1 | 10.8 | 12.3 | 13.2 | 11.0 | 12.9 | 9.2 | 11.7 |
| | range | 6.3–10.9 | 6.6–8.7 | 6.0–9.3 | 6.1–8.5 | 8.2–9.5 | 8.8–10.0 | 8.1–8.7 | 9.9–11.8 | 7.5–9.3 | 9.4–12.8 | 10.4–14.0 | 12.0–15.0 | 7.8–13.0 | 11.0–15.0 | 7.6–10.2 | 10.0–13.2 |

## Mandibular

| | | I₁ | | I₂ | | C | | P₃ | | P₄ | | M₁ | | M₂ | | M₃ | |
|---|---|---|---|---|---|---|---|---|---|---|---|---|---|---|---|---|---|
| | | MD | LL | MD | LL | MD | LL | MD | BL | MD | BL | MD | BL | MD | BL | MD | BL |
| *Au. anamensis* | n | 2 | 1 | 4 | 3 | 7 | 7 | 8 | 8 | 8 | 8 | 9 | 10 | 7 | 7 | 8 | 8 |
| | mean | 6.9 | 7.4 | 7.8 | 8.3 | 10.0 | 10.4 | 12.4 | 9.2 | 9.1 | 11.3 | 12.9 | 12.3 | 14.0 | 13.4 | 15.3 | 13.4 |
| | range | 6.8–6.9 | – | 6.6–8.7 | 7.9–8.6 | 6.6–13.9 | 9.2–11.4 | 11.3–13.4 | 8.6–10.0 | 7.4–9.8 | 9.6–13.2 | 11.6–13.8 | 10.2–14.8 | 13.0–15.9 | 12.3–14.9 | 13.7–17.0 | 12.1–15.2 |
| *Au. afarensis* | n | 7 | 8 | 7 | 6 | 13 | 16 | 27 | 26 | 24 | 21 | 32 | 26 | 31 | 27 | 26 | 23 |
| | mean | 6.7 | 7.1 | 6.7 | 8.0 | 8.8 | 10.4 | 9.6 | 10.6 | 9.8 | 11.0 | 13.1 | 12.6 | 14.3 | 13.4 | 15.3 | 13.5 |
| | range | 5.6–7.7 | – | 5.0–8.0 | 6.7–8.8 | 7.5–11.7 | 8.0–12.4 | 7.9–12.6 | 8.9–13.8 | 7.7–11.4 | 9.8–12.8 | 10.1–14.8 | 11.0–14.0 | 12.1–16.5 | 11.1–15.2 | 13.4–18.1 | 11.3–15.3 |
| *Au. africanus* | n | 11 | 12 | 12 | 13 | 23 | 25 | 20 | 21 | 25 | 23 | 29 | 32 | 38 | 38 | 34 | 35 |
| | mean | 6.2 | 6.7 | 7.2 | 7.9 | 9.4 | 10.1 | 9.7 | 11.5 | 10.4 | 11.6 | 14.0 | 13.0 | 15.7 | 14.5 | 16.3 | 14.6 |
| | range | 4.8–6.9 | 5.7–7.9 | 5.6–8.1 | 6.6–9.2 | 8.5–10.7 | 8.2–12.1 | 8.8–11.0 | 9.9–13.9 | 8.7–12.3 | 9.3–13.2 | 12.4–15.8 | 11.2–15.1 | 14.2–17.7 | 12.8–16.8 | 13.5–18.5 | 12.2–16.8 |
| *Au. sediba* | n | – | 1 | 1 | 1 | 2 | 2 | 1 | 1 | 1 | – | 2 | 2 | 2 | 2 | 2 | 2 |
| | mean | – | 5.9 | 6.6 | 6.6 | 7.7 | 8.0 | 8.1 | 9.2 | 8.8 | 9.7 | 13.1 | 11.4 | 14.5 | 12.8 | 14.9 | 13.2 |
| | range | – | – | – | – | 7.3–8.0 | 7.4–8.6 | – | – | – | – | 13.1–13.1 | 11.3–11.5 | 14.4–14.5 | 12.3–13.2 | 14.9–14.9 | 12.5–13.6 |
| *H. naledi* | n | 7 | 7 | 5 | 6 | 7 | 7 | 9 | 10 | 6 | 6 | 11 | 11 | 9 | 9 | 6 | 5 |
| | mean | 6.1 | 5.4 | 6.9 | 5.9 | 7.1 | 7.1 | 9.0 | 8.8 | 8.7 | 9.1 | 12.2 | 10.7 | 13.3 | 11.2 | 13.4 | 12.1 |
| | range | 5.7–7.0 | 5.3–5.9 | 6.6–7.4 | 5.9–6.0 | 6.4–7.5 | 6.9–7.4 | 8.2–9.4 | 8.2–9.7 | 8.3–9.0 | 8.5–10.2 | 11.3–12.7 | 10.3–11.4 | 12.3–14.0 | 10.7–12.2 | 12.9–13.7 | 11.7–12.8 |
| *H. habilis* | n | 2 | 2 | 2 | 2 | 3 | 2 | 4 | 4 | 3 | 3 | 5 | 5 | 4 | 4 | 4 | 4 |
| | mean | 6.4 | 6.8 | 7.4 | 7.6 | 8.7 | 9.0 | 9.6 | 9.6 | 9.9 | 10.5 | 13.7 | 11.9 | 15.0 | 13.5 | 15.4 | 13.3 |
| | range | 6.4–6.5 | 6.7–7.0 | 7.2–7.7 | 7.6–7.6 | 7.6–9.6 | 7.9–10.0 | 9.0–10.6 | 8.6–11.1 | 9.0–10.5 | 9.9–11.0 | 13.0–14.8 | 10.9–12.8 | 14.2–15.7 | 12.0–15.1 | 14.8–15.9 | 12.4–14.4 |

*Table 2. Continued on next page*

*Table 2. Continued*
**Mandibular**

| | | $I_1$ MD | $I_1$ LL | $I_2$ MD | $I_2$ LL | C MD | C LL | $P_3$ MD | $P_3$ BL | $P_4$ MD | $P_4$ BL | $M_1$ MD | $M_1$ BL | $M_2$ MD | $M_2$ BL | $M_3$ MD | $M_3$ BL |
|---|---|---|---|---|---|---|---|---|---|---|---|---|---|---|---|---|---|
| *H. rudolfensis* | n | – | 1 | – | 1 | – | 1 | 3 | 3 | 6 | 6 | 5 | 5 | 6 | 5 | 3 | 3 |
| | mean | – | 5.4 | – | 6.7 | – | 8.3 | 9.9 | 11.1 | 10.1 | 11.4 | 14.0 | 12.7 | 16.0 | 13.7 | 16.4 | 14.1 |
| | range | – | – | – | – | – | – | 9.0–10.7 | 9.5–12.3 | 8.8–11.8 | 9.8–12.2 | 12.8–15.2 | 11.4–13.2 | 14.0–18.3 | 12.7–14.9 | 15.6–17.0 | 13.1–14.6 |
| *H. erectus* | n | 11 | 12 | 14 | 16 | 14 | 16 | 30 | 30 | 25 | 26 | 43 | 43 | 41 | 40 | 26 | 27 |
| | mean | 6.2 | 6.4 | 7 | 7.2 | 8.7 | 9 | 9 | 10.1 | 8.7 | 10.1 | 12.7 | 11.9 | 13.3 | 12.5 | 12.7 | 11.7 |
| | range | 4.8–7.4 | 5.8–7.1 | 5.3–8.1 | 6.4–8.5 | 7.0–10.3 | 8.0–10.4 | 7.0–12.0 | 8.2–12.0 | 7.2–10.4 | 8.0–12.5 | 9.9–14.8 | 10.1–13.3 | 11.3–15.3 | 10.8–14.3 | 10.0–15.2 | 10.0–14.2 |
| *H. neanderthalensis* | n | 9 | 16 | 23 | 31 | 36 | 41 | 20 | 21 | 23 | 25 | 38 | 40 | 26 | 27 | 18 | 20 |
| | mean | 5.6 | 7.2 | 6.8 | 7.8 | 7.8 | 8.8 | 7.9 | 9.1 | 7.8 | 9.4 | 11.8 | 11.1 | 12.1 | 11.3 | 12.0 | 11.0 |
| | range | 4.2–6.4 | 5.2–8.8 | 5.9–7.5 | 6.8–9.0 | 6.7–8.8 | 6.8–10.3 | 6.6–9.1 | 8.0–10.3 | 6.5–9.4 | 8.5–10.5 | 10.1–13.6 | 10.2–12.9 | 9.3–14.0 | 8.8–12.4 | 11.2–13.9 | 9.9–12.2 |
| *H. heidelbergensis* | n | 21 | 22 | 19 | 20 | 23 | 24 | 22 | 22 | 26 | 26 | 29 | 29 | 29 | 29 | 32 | 32 |
| | mean | 5.6 | 6.7 | 6.5 | 7.3 | 7.6 | 8.7 | 7.9 | 8.9 | 7.2 | 8.7 | 11.3 | 10.6 | 11.2 | 10.5 | 11.5 | 10.0 |
| | range | 4.8–6.5 | 6.0–7.5 | 6.0–7.2 | 6.6–8.0 | 6.9–9.0 | 7.3–10.0 | 7.2–9.0 | 7.6–11.6 | 6.6–8.8 | 7.2–11.7 | 10.4–13.8 | 9.6–13.0 | 9.7–14.6 | 8.5–13.9 | 9.7–13.2 | 8.6–12.5 |
| MP/LP African Homo | n | 5 | 5 | 8 | 8 | 8 | 8 | 8 | 8 | 12 | 9 | 16 | 16 | 20 | 20 | 13 | 13 |
| | mean | 6.0 | 6.8 | 6.8 | 7.2 | 8.8 | 9.6 | 8.6 | 9.8 | 8.6 | 10.3 | 13.1 | 11.8 | 12.5 | 11.7 | 12.4 | 11.5 |
| | range | 5.7–6.4 | 6.1–7.2 | 5.6–8.3 | 6.4–8.0 | 7.8–10.0 | 8.8–10.3 | 7.7–9.0 | 8.6–11.2 | 6.9–9.6 | 9.3–11.4 | 10.7–14.2 | 10.0–13.0 | 10.8–15.0 | 9.2–13.6 | 10.6–13.5 | 9.9–12.7 |

MP, Middle Pleistocene and LP, Late Pleistocene.

further has a flat and squared nasoalveolar clivus, unlike the pronounced maxillary canine juga and prominent pillars of *H. floresiensis*. The mandible of *H. floresiensis* shows a posteriorly inclined post incisive planum with superior and inferior transverse tori, differing from the steeply inclined posterior face of the *H. naledi* mandibular symphysis, which lacks both a post incisive planum or a superior transverse torus. Dentally, *H. naledi* is distinguishable from *H. floresiensis* by the mesiodistal elongation and extensive talonid of the mandibular $P_4$, and the lack of Tomes' root on the mandibular premolars. The molar size gradient of *H. naledi* follows the M1 < M2 < M3 pattern, unlike the M3 < M2 < M1 pattern in *H. floresiensis*, and the mandibular molars are relatively mesiodistally long and buccolingually narrow compared to those of *H. floresiensis*.

*H. naledi* differs from Middle Pleistocene (MP) and Late Pleistocene (LP) *Homo* (here we include specimens sometimes attributed to the putative Early Pleistocene taxon *Homo antecessor*, and MP *Homo heidelbergensis*, *Homo rhodesiensis*, as well as archaic *Homo sapiens* and Neandertals) in exhibiting a smaller cranial capacity. *H. naledi* has its maximum cranial width in the supramastoid region, rather than in the parietal region. It has a clearly defined canine fossa (similar to *H. antecessor*), a shallow anterior palate, and a flat and a squared nasoalveolar clivus. *H. naledi* lacks the bilaterally arched and vertically thickened supraorbital tori found in MP and LP *Homo*. *H. naledi* also differs in exhibiting a root of the zygomatic process of the temporal that is angled downwards approximately 30° relative to FH, a projecting entoglenoid process, a weak vaginal process, a weak crista petrosa, a prominent Eustachian process, a laterally inflated mastoid process, and a small EAM. The *H. naledi* mandible tends to be more gracile than specimens of MP *Homo*. The mandibular canine retains a distinct accessory distal cuspulid not seen in MP and LP *Homo*. Molar cuspal proportions for *H. naledi* do not show the derived reduction of the entoconid and hypoconid that characterizes MP and LP *Homo*. The mandibular $M_3$ is not reduced in DH1, thus revealing an increasing molar size gradient that contrasts with reduction of the $M_3$ in MP and LP *Homo*.

*H. naledi* differs from *H. sapiens* in exhibiting small cranial capacity, a well-defined supraorbital torus and supratoral sulcus, a root of the zygomatic process of the temporal that is angled downwards approximately 30° relative to FH, a large and laterally inflated mastoid with well-developed supramastoid crest, an angular torus, a small vaginal process, a weak crista petrosa, a prominent Eustachian process, a small EAM, a flat and squared nasoalveolar clivus, and a more posteriorly positioned incisive foramen. The *H. naledi* mandible shows a weaker, less well-defined mentum osseum than *H. sapiens*, as well as a slight inferior transverse torus that is absent in humans. The mental foramen is positioned superiorly in *H. naledi*, unlike the mid-corpus height mental foramen of *H. sapiens*. The mandibular canine possesses a distinct accessory distal cuspulid not seen in *H. sapiens*. Molar cuspal proportions for *H. naledi* do not show the derived reduction of the entoconid and hypoconid that characterizes *H. sapiens*. The mandibular $M_3$ is not reduced in *H. naledi*, thus revealing an increasing molar size gradient that contrasts with reduction of the $M_3$ in *H. sapiens*.

## Hand (H1)

*H. naledi* possesses a combination of primitive and derived features not seen in the hand of any other hominin. H1 is differentiated from the estimated intrinsic hand proportions of *Au. afarensis* in having a relatively long thumb ((Mc1 + PP1)/(Mc3 + PP3 + IP3)) (*Rolian and Gordon, 2013*; *Almécija and Alba, 2014*). It is further distinguished from *Au. afarensis*, *Au. africanus,* and *Au. sediba* in having a well-developed crest for both the *opponens pollicis* and first dorsal *interosseous* muscles, a trapezium-scaphoid joint that extends onto the scaphoid tubercle, a relatively large and more palmarly-positioned capitate-trapezoid joint, and/or a saddle-shaped Mc5-hamate joint. *H. naledi* also differs from *Au. sediba* in that it lacks mediolaterally narrow Mc2-5 shafts (*Kivell et al., 2011*). Manual morphology of *Au. garhi* is currently unknown.

H1 is distinguished from *H. habilis* in having a deep proximal palmar fossa with a well-developed ridge distally for the insertion of the *flexor pollicis longus* muscle on the first distal phalanx, and a more proximodistally oriented trapezium-second metacarpal joint. It also differs from both *H. habilis* and *H. floresiensis* by having a relatively large trapezium-scaphoid joint that extends onto the scaphoid tubercle, and from *H. floresiensis* in having a boot-shaped trapezoid with an expanded palmar surface, and a relatively large and more palmarly-positioned capitate-trapezoid joint (*Tocheri et al., 2005*, *2007*; *Orr et al., 2013*).

H1 is dissimilar to hand remains attributed to *Paranthropus robustus*/early *Homo* from Swartkrans (*Susman, 1988*; *Susman et al., 2001*) in having a relatively small Mc1 base and proximal articular

facet, a saddle-shaped Mc5-hamate joint, and more curved proximal and intermediate phalanges of ray 2–5.

Manual morphology of *H. rudolfensis* is currently unknown, and that of *H. erectus* is largely unknown. Still, H1 differs from a third metacarpal attributed to *H. erectus s. l.*, as well as from *Homo neanderthalensis* and *H. sapiens* by lacking a styloid process (*Ward et al., 2013*).

H1 is further distinguished from *H. neanderthalensis* and *H. sapiens* by its relatively small facets for the Mc1 and scaphoid on the trapezium, its low angle between the Mc2 and Mc3 facets on the capitate, and by its long and curved proximal and intermediate phalanges on rays 2–5.

H1 is differentiated from all known hominins in having a Mc1 that combines a mediolaterally narrow proximal end and articular facet with a mediolaterally wide distal shaft and head, a dorsopalmarly flat and strongly asymmetric (with an enlarged palmar-lateral protuberance) Mc1 head, and the combination of an overall later *Homo*-like carpal morphology combined with proximal and intermediate phalanges that are more curved than most australopiths. H1 also differs from all other known hominins except *H. neanderthalensis* in having non-pollical distal phalanges with mediolaterally broad apical tufts (relative to length).

## Femur (U.W. 101-1391)

The femur of *H. naledi* differs from those of all other known hominins in its possession of two well-defined, mediolaterally-running pillars in the femoral neck. The pillars run along the superoanterior and inferoposterior margins of the neck and define a distinct sulcus along its superior aspect.

## Tibia (U.W. 101-484)

The tibia of *H. naledi* differs from those of all other known hominins in its possession of a distinct tubercle for the pes anserinus tendon. The tibia differs from other hominins except *H. habilis*, *H. floresiensis*, and (variably) *H. sapiens* in its possession of a rounded anterior border.

## Foot (F1)

The foot of *H. naledi* differs from the pedal remains of *Au. afarensis, Au. africanus,* and *Au. sediba* in having a calcaneus with a weakly developed peroneal trochlea. F1 also differs from *Au. afarensis* in having a higher orientation of the calcaneal sustentaculum tali. F1 can be further distinguished from pedal remains attributed to *Au. africanus* in having a higher talar head and neck torsion, a narrower Mt1 base, a dorsally expanded Mt1 head, and greater proximolateral to distomedial orientation of the lateral metatarsals. The *H. naledi* foot can be further differentiated from the foot of *Au. sediba* in having a proximodistally flatter talar trochlea, a flat subtalar joint, a diagonally oriented retrotrochlear eminence and a plantar position of the lateral plantar process of the calcaneous, and dorsoplantarly flat articular surface for the cuboid on the Mt4 (*Zipfel et al., 2011*). Pedal remains of *Au. garhi* are currently unknown, and those of *P. robustus* are too poorly known to allow for comparison.

The *H. naledi* foot can be distinguished from the foot of *H. habilis* by the presence of a flatter, non-sloping trochlea with equally elevated medial and lateral margins, a narrower Mt1 base, greater proximolateral to distomedial orientation of the lateral metatarsals, and a metatarsal robusticity ratio of 1 > 5 > 4 > 3 > 2. Pedal morphology in *H. rudolfensis* is currently unknown, and that of *H. erectus* is too poorly known to allow for comparison. The *H. naledi* foot can be distinguished from the foot of *H. floresiensis* by a longer hallux and shorter second through fifth metacarpals relative to hindfoot length, and higher torsion of the talar head and neck.

The foot of *H. naledi* can be distinguished from the foot of *H. sapiens* only by its flatter lateral and medial malleolar facets on the talus, its low angle of plantar declination of the talar head, its lower orientation of the calcaneal sustentaculum tali, and its gracile calcaneal tuber.

## Description

*H. naledi* exhibits anatomical features shared with *Australopithecus*, other features shared with *Homo*, with several features not otherwise known in any hominin species. This anatomical mosaic is reflected in different regions of the skeleton. The morphology of the cranium, mandible, and dentition is mostly consistent with the genus *Homo*, but the brain size of *H. naledi* is within the range of *Australopithecus*. The lower limb is largely *Homo*-like, and the foot and ankle are particularly human in their configuration, but the pelvis appears to be flared markedly like that of *Au. afarensis*. The wrists,

fingertips, and proportions of the fingers are shared mainly with *Homo*, but the proximal and intermediate manual phalanges are markedly curved, even to a greater degree than in any *Australopithecus*. The shoulders are configured largely like those of australopiths. The vertebrae are most similar to Pleistocene members of the genus *Homo*, whereas the ribcage is wide distally like *Au. afarensis*.

*H. naledi* has a range of body mass similar to small-bodied modern human populations, and is similar in estimated stature to both small-bodied humans and the largest known australopiths. We estimated body mass from eight femoral specimens for which subtrochanteric diameters can be measured ('Materials and methods'), with results ranging between 39.7 kg and 55.8 kg (*Table 3*). No femur specimen is sufficiently complete to measure femur length accurately, but the U.W. 101-484 tibia preserves nearly its complete length, allowing a tibia length estimate of 325 mm (*Figure 10*). Estimates for the stature of this individual based on African human population samples range between 144.5 and 147.8 mm. Again, this stature estimate is similar to small-bodied modern human populations. It is within the range estimated for Dmanisi postcranial elements (*Lordkipanidze et al., 2007*), and slightly smaller than estimated for early *Homo* femoral specimens KNM-ER 1472 and KNM-ER 1481. Some large australopiths also had long tibiae and presumably comparably tall statures, as evidenced by the KSD-VP 1/1 skeleton from Woranso-Mille (*Haile-Selassie et al., 2010*).

The endocranial volume of all *H. naledi* specimens is clearly small compared to most known examples of *Homo*. We combined information from the most complete cranial vault specimens to arrive at an estimate of endocranial volume for both larger (presumably male) and smaller (presumably female) individuals (larger composite depicted in *Figure 11*). The resulting estimates of approximately 560cc and 465cc, respectively, overlap entirely with the range of endocranial volumes known for australopiths. Within the genus *Homo*, only the smallest specimens of *H. habilis,* one single *H. erectus* specimen, and *H. floresiensis* overlap with these values.

Despite its small vault size, the cranium of *H. naledi* is structurally similar to those of early *Homo*. Frontal bossing is evident, as is a marked degree of parietal bossing. There is no indication of metopic keeling, though there is slight parasagittal keeling between bregma and lambda, and some prelambdoidal flattening. The cranial vault bones are generally thin, becoming somewhat thicker in the occipital region. The supraorbital torus is well developed, though weakly arched, and is bounded posteriorly by a well-developed supratoral sulcus. The lateral corners of the supraorbital torus are rounded and relatively thin. The temporal lines are widely spaced, and there is no indication of a sagittal crest or temporal/nuchal cresting. The temporal crest is positioned on the posterior aspect of the lateral supraorbital torus, rather than on the superior aspect as in australopiths. At the posteroinferior extent of the temporal lines, they curve anteroinferiorly presenting a well-developed angular torus. The crania have a pentagonal outline in posterior view, with the greatest vault breadth located in the supramastoid region. The nuchal region exhibits sexually dimorphic development of nuchal muscle markings and the external occipital protuberance, and there is a clear indication of

**Table 3**. Dinaledi body mass estimates from femur specimens preserving subtrochanteric diameters

| Specimen ID | Side | AP subtrochanteric breadth | ML subtrochanteric breadth | Mass (a) | Mass (b) |
|---|---|---|---|---|---|
| U.W. 101-002 | R | 18.5 | 23.6 | 40.0 | 44.7 |
| U.W. 101-003 | R | 21.6 | 31.4 | 54.5 | 55.8 |
| U.W. 101-018 | R | 18.1 | 23.8 | 39.7 | 44.4 |
| U.W. 101-226 | L | 19.1 | 24.0 | 41.3 | 45.7 |
| U.W. 101-1136 | R | 16.9 | 25.5 | 39.7 | 44.4 |
| U.W. 101-1391 | R | 18.8 | 23.9 | 40.8 | 45.3 |
| U.W. 101-1475 | L | 18.8 | 29.0 | 46.5 | 49.7 |
| U.W. 101-1482 | L | 20.7 | 28.9 | 49.7 | 52.1 |

Regression equations described in 'Materials and methods'. Mass (a) based on forensic statures from European individuals. Mass (b) based on multiple population sample. The two estimates diverge somewhat for smaller femora.

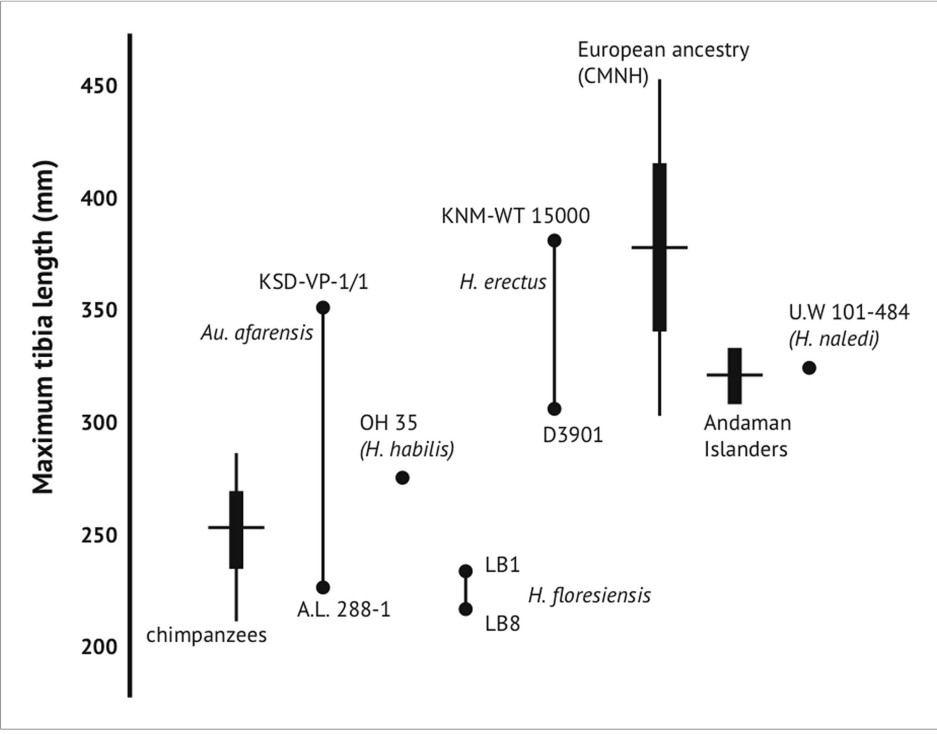

**Figure 10**. Maximum tibia length in *H. naledi* and other hominins. Maximum tibia length for U.W. 101-484, compared to other nearly complete hominin tibia specimens. *Australopithecus afarensis* represented by A.L. 288-1 and KSD-VP-1/1 (*Haile-Selassie et al., 2010*); *Homo erectus* represented by D3901 from Dmanisi and KNM-WT 15000; *Homo habilis* by OH 35; *Homo floresiensis* by LB1 and LB8 (*Brown et al., 2004*; *Morwood et al., 2005*). Chimpanzee and contemporary European ancestry humans from Cleveland Museum of Natural History (*Lee, 2001*); Andaman Islanders from *Stock (2013)*. Vertical lines represent sample ranges; bars represent 1 standard deviation.

a tuberculum linearum in addition to the external occipital protuberance. In superior view the vault tapers from posterior to anterior, though post-orbital constriction is slight. The squamosal suture is low and gently curved, and parietal striae are well defined. The lateral margins of the orbits face laterally. A small zygomaticofacial foramen is typically present near the center of the zygomatic bone. The root of the zygomatic process of the maxilla is anteriorly positioned, at the level of the $P^3$ or the $P^4$. There is no indication of a zygomatic prominence, and the zygomatic arches do not flare laterally to any extent. The root of the zygomatic process of the temporal is angled downwards approximately 30° relative to FH. The root of the zygomatic process of the temporal begins to laterally expand above the level of the mandibular fossa, rather than above the level of the EAM as in australopiths. The mandibular fossa is somewhat large, and moderately deep. The articular eminence of the mandibular fossa is saddle-shaped, and oriented posteroinferiorly. Almost the entire mandibular fossa is positioned medial to the temporal squama. The entoglenoid process is elongated and faces primarily laterally. The postglenoid process is small and closely appressed to the tympanic, forming part of the posterior wall of the fossa. The petrotympanic is distinctly coronally oriented. The vaginal process is small but distinct. The crista petrosa is weakly developed and not notably sharpened. There is a strong Eustachian process. The external auditory meatus is small, oval-shaped, and obliquely oriented, and a distinct suprameatal spine is present. The mastoid region is slightly laterally inflated. The mastoid process is triangular in cross-section, with a rounded apex and a mastoid crest. The digastric groove is deep and narrow, alongside a marked juxtamastoid eminence. The canine juga are weakly developed and there is no indication that anterior pillars would have been present. A shallow, ill-defined canine fossa is indicated. The nasoalveolar clivus is flat and square-shaped. The parabolic-shaped palate is broad and anteriorly shallow, becoming deeper posteriorly.

The mandibular dentition of *H. naledi* is arranged in a parabolic arch. The alveolar and basal margins of the corpus diverge slightly. A single, posteriorly opening mental foramen is positioned slightly above

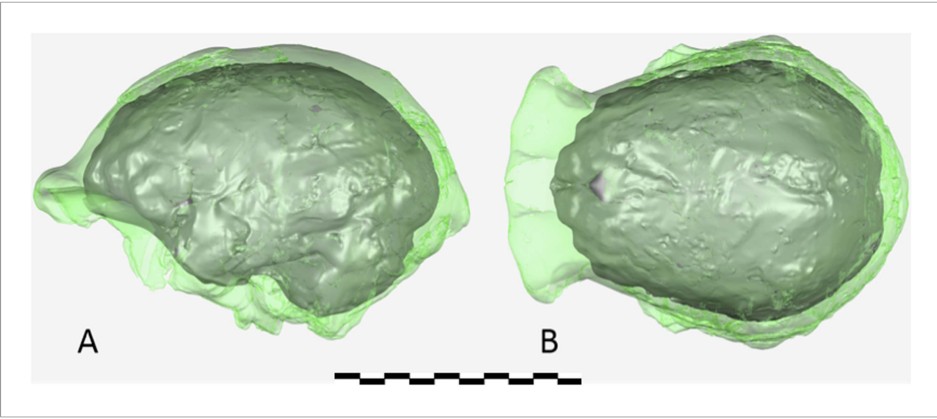

**Figure 11**. Virtual reconstruction of the endocranium of the larger composite cranium from DH1 and DH2 overlaid with the ectocranial surfaces. (**A**) Lateral view. (**B**) Superior view. The resulting estimate of endocranial volume is 560cc. Scale bar = 10 cm.

the mid-corpus level, between the position of the $P_3$ and the $P_4$. The mandibular corpus is relatively gracile, with a well-developed lateral prominence whose maximum extent is typically at the $M_2$. A slight supreme lateral torus (of Dart) weakly delineates the extramolar sulcus from the lateral corpus. The superior lateral torus is moderately developed, running anteriorly to the mental foramen where it turns up to reach the $P_3$ jugum. The marginal torus is moderately developed, and defines a moderate intertoral sulcus. The posterior and anterior marginal tubercles are indicated only as slight roughenings of bone. The gracile mandibular symphysis is vertically oriented. A well-developed mental protuberance and weak lateral tubercles are delineated by a slight mandibular incisure, thereby presenting a weak mentum osseum. The post-incisive planum is steeply inclined and not-shelf-like. There is no superior transverse torus, while a weak, basally oriented inferior transverse torus is present. The anterior and posterior subalveolar fossae are continuous and deep, overhung by a well-developed alveolar prominence. The extramolar sulcus is moderately wide. The root of the ramus of the mandible originates high on the corpus at the level of the $M_2$. Strong ectoangular tuberosities are indicated. A large mandibular foramen is present, with a diffusely defined mylohyoid groove.

Like the skull, the dentition of *H. naledi* compares most favorably to early *Homo* samples. Yet compared to samples of *H. habilis*, *H. rudolfensis*, and *H. erectus*, the teeth of *H. naledi* are comparatively quite small, similar in dimensions to much later samples of *Homo*. With both small post-canine teeth and a small endocranial volume, *H. naledi* joins *Au. sediba* and *H. floresiensis* in an area distinct from the general hominin relation of smaller post-canine teeth in species with larger brains (*Figure 12*).

In comparison to *H. habilis*, *H. rudolfensis*, and *H. erectus*, the teeth of *H. naledi* are not only small, but also markedly simple in crown morphology. Maxillary and mandibular molars lack extensive crenulation, secondary fissures and supernumerary cusps. The $M^1$ has an equal-sized metacone and paracone, and has a slight expression of Carabelli's trait represented by a small cusp or shallow pit. $I^1$ exhibits slight occlusal curvature with trace marginal ridges and variably small *tuberculum dentale*. $I^2$ exhibits greater occlusal curvature and *tuberculum dentale* expression but neither upper incisor has double shovelling or interruption groove. The mandibular canines of *H. naledi* have a small occlusal area, and have a distal marginal cuspule as a topographically distinct expression of the cingular margin. The $P_3$ is double-rooted, fully bicuspid with metaconid and protoconid of approximately equal height and occlusal area separated by a distinct longitudinal groove, has a distally extensive talonid, and an occlusal outline approximately symmetrical with respect to the mesiodistal axis. $P_4$ likewise has a distally extensive talonid and approximately symmetrical occlusal outline (*Figure 5*). $M_1$ and $M_2$ lack cusp 6 and cusp 7, except for very slight expression in a small fraction of specimens, and have a very faint subvertical depression rather than a distinct or extensive protostylid. Like australopiths and some early *Homo* specimens, *H. naledi* has an increasing molar size gradient in the mandibular dentition (M1 < M2 < M3).

The lower limb of *H. naledi* is defined not only by a unique combination of primitive and derived traits, but also by the presence of unique features in the femur and tibia. Like all other bipedal

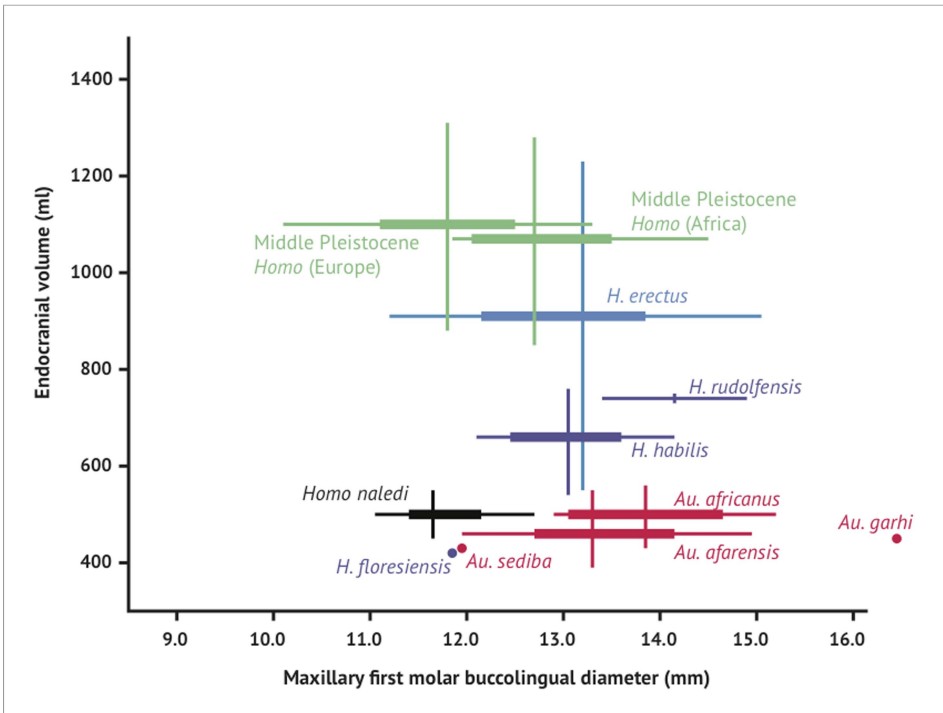

**Figure 12**. Brain size and tooth size in hominins. The buccolingual breadth of the first maxillary molar is shown here in comparison to endocranial volume for many hominin species. *H. naledi* occupies a position with relatively small molar size (comparable to later *Homo*) and relatively small endocranial volume (comparable to australopiths). The range of variation within the Dinaledi sample is also fairly small, in particular in comparison to the extensive range of variation within the *H. erectus sensu lato*. Vertical lines represent the range of endocranial volume estimates known for each taxon; each vertical line meets the horizontal line representing M$^1$ BL diameter at the mean for each taxon. Ranges are illustrated here instead of data points because the ranges of endocranial volume in several species are established by specimens that do not preserve first maxillary molars.

hominins, *H. naledi* possesses a valgus knee and varus ankle. The femoral neck is long, anteverted, and anteroposteriorly compressed. Muscle insertions for the *M. gluteus maximus* are strong and the femur has a well-marked linea aspera with pilaster variably present. The patella is relatively anteroposteriorly thick. The tibia is mediolaterally compressed with a rounded anterior border, a large proximal attachment for the *M. tibialis posterior*, and a thin medial malleolus. The fibula is gracile with laterally oriented lateral malleolus, a relatively circular neck and a convex surface for the proximal attachment of the *M. peroneus longus*. Unique features in the lower limb of *H. naledi* include a depression in the superior aspect of the femoral neck that results in two mediolaterally oriented pillars inferoposteriorly and superoanteriorly, and a strong distal attachment of the pes anserinus on the tibia.

The foot and ankle of *H. naledi* are largely humanlike (*Figure 9*). The tibia stands orthogonally upon the talus, which is moderately wedged, with a mediolaterally flat trochlea having medial and lateral margins at even height, a form distinct from the strong keeling seen in OH 8 (*H. habilis*) and several tali from Koobi Fora. The talar head and neck exhibit strong, humanlike torsion; the horizontal angle is higher than in most humans, similar to that found in australopiths. The calcaneus is only moderately robust, but possesses the plantar declination of the retrotrochlear eminence and plantarly positioned lateral plantar process found in both modern humans and *Au. afarensis*. The peroneal trochlea is small, unlike that found in australopiths and similar only to that in *H. sapiens* and Neanderthals. The talonavicular, subtalar joints and calcaneocuboid joints are humanlike in possessing minimal ranges of motion and evidence for a locking, rigid midfoot. The intermediate and lateral cuneiforms are proximodistally elongated. The hallucal tarsometatarsal joint is flat and proximodistally aligned indicating that *H. naledi* possessed an adducted, non-grasping hallux. The head of the first metatarsal is mediolaterally expanded dorsally, indicative of

a humanlike windlass mechanism. The foot possesses humanlike metatarsal lengths, head proportions, and torsion.

The phalanges are moderately curved, slightly more so than in *H. sapiens*. The only primitive anatomies found in the foot of *H. naledi* are the talar head and neck declination and sustentaculum tali angles, suggestive of a lower arched foot with a more plantarly positioned and horizontally inclined medial column than typically found in modern humans (*Harcourt-Smith et al., 2015*).

The axial skeleton presents a combination of derived (mainly aspects of the vertebrae) and seemingly primitive (mainly the ribs) traits. The preserved adult T10 and T11 vertebrae are proportioned similarly to Pleistocene *Homo*, with transverse process morphology most similar to Neandertals. The neural canals of these vertebrae are large in comparison to the diminutive overall size of the vertebrae, proportionally recalling Dmanisi *H. erectus*, Neandertals, and modern humans. The 11th rib is straight (uncurved), similar to *Au. afarensis*, and the shape of the upper rib cage appears narrow, as assessed from first and second rib fragments, suggesting that the thorax was pyramidal in shape. The 12th rib presents a robust shaft cross-section most similar to Neandertals. This combination is not found in other hominins and might reflect allometric scaling at a small trunk size.

The Dinaledi iliac blade is flared and shortened anteroposteriorly, resembling *Au. afarensis* or *Au. africanus.* The ischium is short with a narrow tuberoacetabular sulcus, and the ischiopubic and iliopubic rami are thick, resembling *Au. sediba* and *H. erectus*. This combination of iliac and ischiopubic features has not been found in other fossil hominins (*Figure 13*).

The shoulder of *H. naledi* is configured with the scapula situated high and lateral on the thorax, short clavicles, and little or no torsion of the humerus. The humerus is notably slender for its length, with prominent greater and lesser tubercles bounding a deep bicipital groove, with a small, non-projecting humeral deltoid tuberosity and brachioradialis crest. Distally, the humerus has a wide lateral distodorsal pillar and narrow medial distodorsal pillar, and a medially-displaced olecranon fossa with septal aperture. The Dinaledi radius and ulna diaphyses exhibit little curvature. The radius has a globular radial tuberosity, prominent pronator quadratus crest, and reduced styloid process.

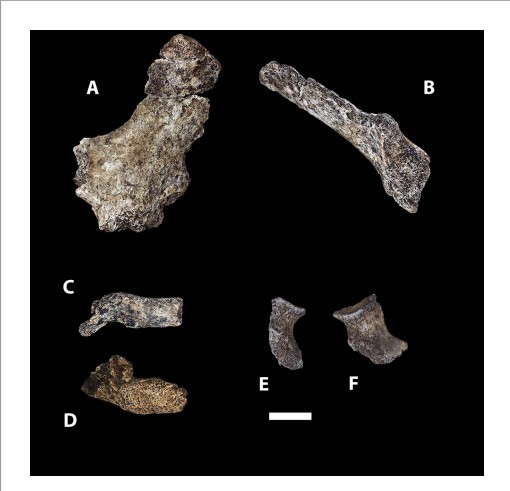

**Figure 13**. Selected pelvic specimens of *H. naledi*. U.W. 101-1100 ilium in (**A**) lateral view showing a weak iliac pillar relatively near the anterior edge of the ilium, with no cristal tubercle development; (**B**) anterior view, angled to demonstrate the degree of flare, which is clear in comparison to the subarcuate surface. U.W. 101-723 immature sacrum in (**C**) anterior view; and (**D**) superior view. U.W. 101-1112 ischium in (**E**) lateral view; and (**F**) anterior view, demonstrating relatively short tuberacetabular diameter. Scale bar = 2 cm.

The hand shares many derived features of modern humans and Neandertals in the thumb, wrist, and palm, but has relatively long and markedly curved fingers (*Kivell et al., 2015*). The thumb is long relative to the length of the other digits, and includes a robust metacarpal with well-developed intrinsic (*M. opponens pollicis* and *M. first dorsal interosseous*) muscle attachments (*Figure 6*). The pollical distal phalanx is large and robust with a well-developed ridge along the distal border of a deep proximal palmar fossa for the attachment of *flexor pollicis longus* tendon. Ungual spines also project proximopalmarly from a radio-ulnarly expanded apical tuft with a distinct area for the ungual fossa. The wrist includes a boot-shaped trapezoid with an expanded non-articular palmar surface, an enlarged and palmarly-expanded trapezoid-capitate joint, and a trapezium-scaphoid joint that extends further onto the scaphoid tubercle. Overall, carpal shapes and articular configurations are very similar to those of modern humans and Neandertals, and unlike those of great apes and other extinct hominins. However, the *H. naledi* wrist lacks a third metacarpal styloid process, has a more radioulnarly oriented capitate-Mc2 joint, and has a relatively small trapezium-Mc1 joint compared to humans and Neandertals. Moreover, the phalanges are long (relative to the palm) and more curved than most australopiths.

## Discussion

The overall morphology of *H. naledi* places it within the genus *Homo* rather than *Australopithecus* or other early hominin genera. The shared derived features that connect *H. naledi* with other members of *Homo* occupy most regions of the *H. naledi* skeleton and represent distinct functional systems, including locomotion, manipulation, and mastication. Locomotor traits shared with *Homo* include the absolutely long lower limb, with well-marked linea aspera, strong *M. gluteus maximus* insertions, gracile fibula and generally humanlike ankle and foot. These aspects of the lower limb suggest enhanced locomotor performance for a striding gait. The *H. naledi* hand shares aspects of *Homo* morphology in the wrist, thumb and palm, pointing to enhanced object manipulation ability relative to australopiths, including *Au. sediba* (*Kivell et al., 2011*; *Kivell et al., 2015*). *H. naledi* lacks the powerful mastication that typifies *Australopithecus* and *Paranthropus*, with generally small teeth across the dentition, gracile mandibular corpus and symphysis, laterally-positioned temporal lines, slight postorbital constriction and non-flaring zygomatic arches. The upper limb, shoulder and ribcage have a more primitive morphological pattern, but do not preclude affiliating *H. naledi* with *Homo*, particularly considering that postcranial remains of *H. habilis* appear to reflect an australopith-like body plan (*Johanson et al., 1986*). Locomotor, manipulatory, and masticatory systems have both historical and current importance in defining *Homo* (*Wood and Collard, 1999*; *Holliday, 2012*; *Antón et al., 2014*), and *H. naledi* fits within our genus in these respects.

The structural configuration of the *H. naledi* cranium, beyond the functional aspects of mastication, is likewise shared with *Homo*. As in many specimens of *H. erectus* and *H. habilis*, the *H. naledi* vault includes a well-developed and moderately arched supraorbital torus, marked from the frontal squama by a continuous supratoral sulcus, frontal bossing. Further, as in many *H. erectus* crania, *H. naledi* exhibits a marked angular torus and occipital torus. The *H. naledi* face includes a flat and squared nasoalveolar clivus, comparable to *H. rudolfensis* (*Leakey et al., 2012*), and weak canine fossae. While its anatomy places it unambiguously within *Homo*, the *H. naledi* cranium and dentition lack many derived features shared by MP and LP *Homo* and *H. sapiens*. The australopith-like features of the postcranium, including the ribcage, shoulder, proximal femur, and relatively long, curved fingers, also depart sharply from the morphology present in MP humans and *H. sapiens*. The similarities of *H. naledi* to earlier members of *Homo*, including *H. habilis*, *H. rudolfensis*, and *H. erectus*, suggest that this species may be rooted within the initial origin and diversification of our genus.

The fossil record of early *Homo* and *Homo*-like australopiths has rapidly increased during the last 15 years, and this accumulating evidence has changed our perspective on the rise of our genus. Many skeletal and behavioral features observed to separate later *Homo* from earlier hominins were formerly argued to have arisen as a single adaptive package, including increased brain size, tool manipulation, increased body size, smaller dentition, and greater commitment to terrestrial long-distance walking or running (*Wood and Collard, 1999*; *Hawks et al., 2000*). But we now recognize that such features appeared in different combinations in different fossil samples (*Antón et al., 2014*). The Dmanisi postcranial sample (*Lordkipanidze et al., 2007*) and additional cranial remains of *H. erectus* from Dmanisi (*Gabunia et al., 2000*; *Vekua et al., 2002*; *Lordkipanidze et al., 2013*) and East Africa (*Spoor et al., 2007*; *Leakey et al., 2012*), demonstrate that larger brain size and body size did not arise synchronously with improved locomotor efficiency and adaptations to long-distance walking or running in *H. erectus* (*Holliday, 2012*; *Antón et al., 2014*). Further, the discovery of *Au. sediba* showed that a mosaic of *Homo*-like hand, pelvis and aspects of craniodental morphology can occur within a species with primitive body size, limb proportions, lower limb and foot morphology, thorax shape, vertebral morphology, and brain size (*Berger et al., 2010*; *Carlson et al., 2011*; *Kivell et al., 2011*; *Churchill et al., 2013*; *DeSilva et al., 2013*; *Schmid et al., 2013*). *H. naledi* presents yet a different combination of traits. This species combines a humanlike body size and stature with an australopith-sized brain; features of the shoulder and hand apparently well-suited for climbing with humanlike hand and wrist adaptations for manipulation; australopith-like hip mechanics with humanlike terrestrial adaptations of the foot and lower limb; small dentition with primitive dental proportions. In light of this evidence from complete skeletal samples, we must abandon the expectation that any small fragment of the anatomy can provide singular insight about the evolutionary relationships of fossil hominins.

A recent phylogenetic analysis of fossil hominins based on craniodental morphology placed *Au. sediba* at the base of the genus *Homo* (*Dembo et al., 2015*), in agreement with earlier analyses of

this species (*Berger et al., 2010*). The cranial and dental affinities identified between *Au. sediba* and *Homo* include many features shared by *H. naledi*. But *H. naledi* and *Au. sediba* share different postcranial features with other species of *Homo*. Resolving the phylogenetic placement of *H. naledi* will require both postcranial and craniodental evidence to be integrated together. Such integration poses a challenge because of the poor representation of several key species both within and outside of *Homo*, most notably *H. habilis*, for which postcranial evidence is slight, and *H. rudolfensis* for which no associated postcranial remains are known. We propose the testable hypothesis that the common ancestor of *H. naledi*, *H. erectus*, and *H. sapiens* shared humanlike manipulatory capabilities and terrestrial bipedality, with hands and feet like *H. naledi*, an australopith-like pelvis and the *H. erectus*-like aspects of cranial morphology that are found in *H. naledi*. Enlarged brain size was evidently not a necessary prerequisite for the generally human-like aspects of manipulatory, locomotor, and masticatory morphology of *H. naledi*.

Although it contains an unprecedented wealth of anatomical information, the Dinaledi deposit remains undated (*Dirks et al., 2015*). Considering that *H. naledi* is a morphologically primitive species within our genus, an age may help elucidate the ecological circumstances within which *Homo* arose and diversified. If the fossils prove to be substantially older than 2 million years, *H. naledi* would be the earliest example of our genus that is more than a single isolated fragment. The sample would illustrate a model for the relation of adaptive features of the cranium, dentition and postcranium during a critical time interval that is underrepresented by fossil evidence of comparable completeness. A date younger than 1 million years ago would demonstrate the coexistence of multiple *Homo* morphs in Africa, including this small-brained form, into the later periods of human evolution. The persistence of such a species with clear adaptations for manipulation and grip, alongside MP humans or perhaps even alongside modern humans, would challenge many assumptions about the development of the archaeological record in Africa.

The depth of evidence of *H. naledi* may provide a perspective on the variation to be expected within fossil hominin taxa (*Lordkipanidze et al., 2013*; *Bermúdez de Castro et al., 2014*). The entire Dinaledi collection is remarkably homogeneous. There is very little size variation among adult elements within the collection. Eight body mass estimates from the femur (*Table 2*) have a standard deviation of only 4.3 kilograms, for a body mass coefficient of variation (CV) of only 9%. The CV of body mass within most human populations is substantially higher than this, with an average near 15% (*McKellar and Hendry, 2009*). Likewise, the size variation of cranial and dental elements is minimal. With 11 mandibular first molars, the CV of buccolingual breadth is only 3.2% and for 13 maxillary first molars the CV of buccolingual breadth is only 2.0% (buccolingual breadth is used because it is not subject to variance from interproximal wear). Not only size, but also anatomical shape and form are homogeneous within the sample. Almost every aspect of the morphology of the dentition, including the distinctive form of the lower premolars, the distal accessory cuspule of the mandibular canines, and the expression of nonmetric features that normally vary in human populations, is uniform in every specimen from the collection. The distinctive aspects of cranial morphology are repeated in every specimen, and even the aspects that normally vary among individuals of different body size or between sexes exhibit only slight variation among the Dinaledi remains. One of the most unique aspects of *H. naledi* is the morphology of the first metacarpal; the derived aspects of this anatomy are present in every one of seven first metacarpal specimens in the collection (*Figure 14*). Unlike any other fossil hominin site in Africa, the Dinaledi Chamber seems to preserve a large number of individuals from a single population, one with variation equal to or less than that found within local populations of modern humans.

The Dinaledi collection is the richest assemblage of associated fossil hominins ever discovered in Africa, and aside from the Sima de los Huesos collection and later Neanderthal and modern human samples, it has the most comprehensive representation of skeletal elements across the lifespan, and from multiple individuals, in the hominin fossil record. The abundance of evidence from this assemblage supports our emerging understanding that the genus *Homo* encompassed a variety of evolutionary experiments (*Antón et al., 2014*), with diversity now evident for fossil *Homo* in each of the few intensively explored parts of Africa (*Leakey et al., 2012*). But as much as it advances our knowledge, *H. naledi* also highlights our ignorance about ancient *Homo* across the vast geographic span of the African continent. The tree of *Homo*-like hominins is far from complete: we have missed key transitional forms and lineages that persisted for hundreds of thousands of years. With an increasing pace of discovery from the field and the laboratory, more light will be thrown on the origin of humans.

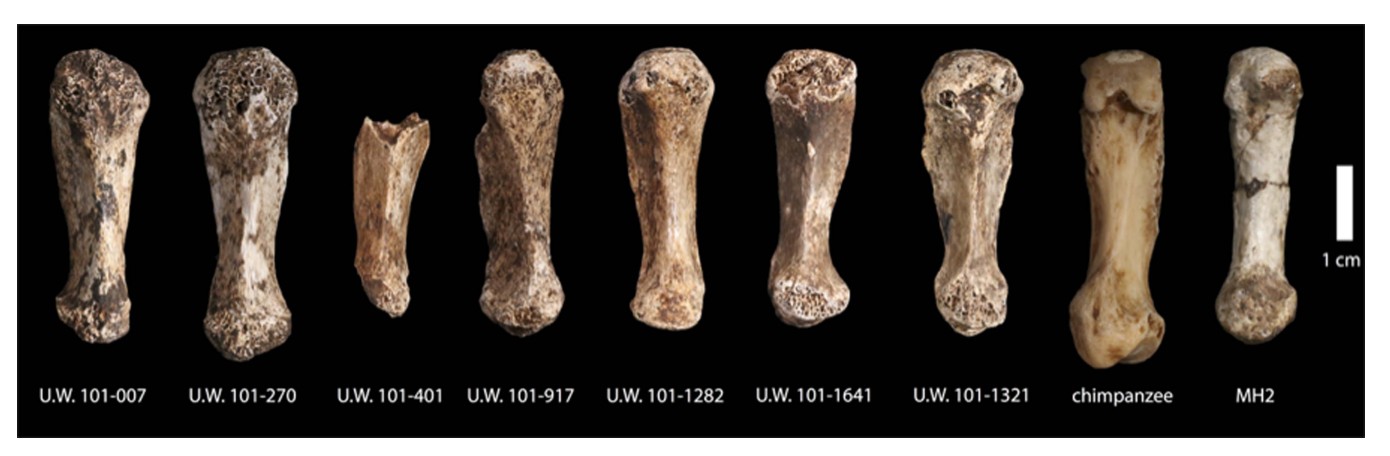

**Figure 14**. First metacarpals of *H. naledi*. Seven first metacarpals have been recovered from the Dinaledi Chamber. U.W. 101-1321 is the right first metacarpal of the associated Hand 1 found in articulation. U.W. 101-1282 and U.W. 101-1641 are anatomically similar left and right first metacarpals, which we hypothesize as antimeres, both were recovered from excavation. U.W. 101-007 was collected from the surface of the chamber, and exhibits the same distinctive morphological characteristics as all the first metacarpals in the assemblage. All of these show a marked robusticity of the distal half of the bone, a very narrow, 'waisted' appearance to the proximal shaft and proximal articular surface, prominent crests for attachment of *M. opponens pollicis* and *M. first dorsal interosseous*, and a prominent ridge running down the palmar aspect of the bone. The heads of these metacarpals are dorsopalmarly flat and strongly asymmetric, with an enlarged palmar-radial protuberance. These distinctive features are present among all the first metacarpals in the Dinaledi collection, and are absent from any other hominin sample. Their derived nature is evident in comparison to apes and other early hominins, here illustrated with a chimpanzee first metacarpal and the MH2 first metacarpal of *Australopithecus sediba*.

## Materials and methods

### Comparative hominin specimens examined in this study

In the differential diagnosis of *H. naledi*, we have compared the holotype DH1, paratypes, and other referred material to fossil evidence from previously-identified hominin taxa. Our goal is to provide a diagnosis for *H. naledi* that is clear in reference to widely recognized hominin hypodigms. Different specialists continue to disagree about the composition and anatomical breadth represented by these hominin taxa and attribution of particular specimens to them (see e.g., *Wood and Collard, 1999*; *Lordkipanidze et al., 2013*; *Antón et al., 2014* on early *Homo* taxa). We do not intend to take any position on such disagreements by our selection of comparative samples for *H. naledi*.

We have been cautious in our attribution of postcranial specimens to hominin taxa, particularly in the African Plio-Pleistocene, where it has been demonstrated multiple hominin taxa coexisted in time, if not in geographical space. Because the purpose of this study is differential diagnosis in reference to known taxa, unattributed specimens are not germane, although in certain cases there are well-accepted attributions to genus for specimens (e.g., *Homo* sp. or *Australopithecus* sp.) as cited below. We have included some specimens in comparisons because they are relatively complete, even if they cannot be attributed to a species, because few hominin taxa are represented by evidence across the entire skeleton. For some anatomical characters, parts are preserved only for MP or later hominin samples, so we have included such comparisons to make clear how *H. naledi* compares in these elements to the (few) known fossil examples.

This study relies upon observations and measurements taken from original fossils by the authors, observations taken from casts, and observations taken from the literature. These observations are in large part standard anatomical practice; where features are specially described in previous studies we have referenced those here. For this study, a cast collection was assembled including the Phillip V. Tobias research collection at the University of the Witwatersrand and loans of cast materials from the University of Wisconsin–Madison, University of Michigan, American Museum of Natural History, New York University, University of Colorado–Denver, University of Delaware, Texas A&M University, and the personal collections of Peter Schmid, Milford Wolpoff and Rob Blumenschine. We extend our gratitude to the curators of fossil

collections and the generosity of these institutions in facilitating this research, both in South Africa and throughout the world.

This list of skeletal materials extends the list of craniodental comparative material used in diagnosing *Au. sediba*, with many of the hypodigms identical to that study (*Berger et al., 2010*). Where we have had first-hand access to original specimens, we rely upon our own observations; we therefore do not refer readers to other sources for these data.

### Australopithecus afarensis
The samples attributed to *Au. afarensis* from Hadar, Laetoli, the Middle Awash, Woranso-Mille and Dikika were utilized. For this taxon we relied upon published reports (*Johanson et al., 1982*; *Kimbel et al., 2004*; *Drapeau et al., 2005*; *Alemseged et al., 2006*; *Haile-Selassie et al., 2010*; *Ward et al., 2012*), in addition to our own observations on original fossils and casts.

### Australopithecus africanus
The samples attributed to *Au. africanus* from Taung, Sterkfontein and Makapansgat were employed. Original specimens were examined first-hand by the authors.

### Australopithecus garhi
The cranium BOU-VP-12/130 from Bouri was included, with data taken from a published report (*Asfaw et al., 1999*).

### Australopithecus sediba
The partial skeletons MH1 and MH2 from Malapa, South Africa were included in this study, based on examination of the original specimens by the authors.

### Paranthropus aethiopicus
The cranium KNM-WT 17000 was examined first-hand for this study.

### Paranthropus boisei
Samples from the Omo Shungura sequence, East Lake Turkana, Olduvai Gorge and Konso were included in this study. Original specimens from Olduvai Gorge and East Lake Turkana were examined first-hand, while casts and published reports (*Tobias, 1967*; *Suwa et al., 1996*, *1997*; *Domínguez-Rodrigo et al., 2013*) were used to study the Omo and Konso materials. Our postcranial considerations of *P. boisei* are very limited and we did not rely upon the association of KNM-ER 1500 (*Grausz et al., 1988*) to derive information about the postcranial skeleton of *P. boisei*.

### Paranthropus robustus
The samples from Kromdraai, Swartkrans, Sterkfontein, Drimolen, Gondolin, and Coopers were included in this study. First-hand observations of original specimens from all localities were used with the exception of Drimolen fossils, which were compared using published reports (*Keyser, 2000*; *Keyser et al., 2000*).

### Homo habilis
Samples from Olduvai Gorge, East Lake Turkana, the Omo Shungura sequence, Hadar, and Sterkfontein were included in this study. Original Olduvai Gorge and East Lake Turkana fossils were examined first-hand, while for the Omo and Hadar materials we relied on our original observations on casts and originals and published reports (*Boaz and Howell, 1977*; *Tobias, 1991*; *Kimbel et al., 1997*). We include the following fossils in the hypodigm of *H. habilis*: A.L. 666-1, KNM-ER 1478, KNM-ER 1501, KNM-ER 1502, KNM-ER 1805, KNM-ER 1813, KNM-ER 3735, OH 4, OH 6, OH 7, OH 8, OH 13, OH 15, OH 16, OH 21, OH 24, OH 27, OH 31, OH 35, OH 37, OH 39, OH 42, OH 44, OH 45, OH 62, OMO-L894-1, and Stw 53. We recognize that some authors (including some of the authors of this paper) prefer to classify OH 62, Stw 53 and A.L. 666-1 outside of *H. habilis*, (e.g., as *Homo gautengensis* which we do not recognize as valid), or even outside the genus *Homo*; these specimens expand the morphological and temporal variability encompassed within *H. habilis*.

### Homo rudolfensis
Samples from Olduvai Gorge, East Lake Turkana, and Lake Malawi were included in this study. The East Lake Turkana fossils available prior to 2010 were examined first-hand, while for the Olduvai and Lake Malawi fossils and KNM-ER 60000, 62000, and 62003 we relied on original observations on fossils and casts as well as published reports (*Schrenk et al., 1993*; *Blumenschine et al., 2003*; *Leakey et al., 2012*). We include the following fossils in the hypodigm of *H. rudolfensis*: KNM-ER 819,

KNM-ER 1470, KNM-ER 1482, KNM-ER 1483, KNM-ER 1590, KNM-ER 1801, KNM-ER 1802, KNM-ER 3732, KNM-ER 3891, KNM-ER 60000, KNM-ER 62000, KNM-ER 62003, OH 65, and UR 501. We do recognize that KNM-ER 60000 and KNM-ER 1802 present some conflicting anatomy that some authors have argued precludes them as conspecific specimens (*Leakey et al., 2012*); by considering both, we aim to be conservative as they encompass more variation within *H. rudolfensis*.

### Homo erectus

Samples from Buia, Chemeron, Daka, Dmanisi, East and West Lake Turkana, Gona, Hexian, Konso, Mojokerto, Olduvai Gorge, Sangiran, Swartkrans, Trinil, and Zhoukoudian were included in this study. South African material is of special interest in this comparison because of the geographic proximity, and because of the difficulty of clearly identifying *Homo* specimens within the large fossil sample from Swartkrans. In particular, the following specimens from Swartkrans are considered to represent *H. erectus*: SK 15, SK 18a, SK 27, SK 43, SK 45, SK 68, SK 847, SK 878, SK 2635, SKW 3114, SKX 257/258, SKX 267/2671, SKX 268, SKX 269, SKX 334, SKX 339, SKX 610, SKX 1756, SKX 2354, SKX 2355, SKX 2356, and SKX 21204. It has been suggested (*Grine et al., 1993*, *1996*) that SK 847 and Stw 53 might represent the same taxon, and that this taxon is a currently undiagnosed species of *Homo* in South Africa. However, we agree with *Clarke (1977*; *2008)* that SK 847 can be attributed to *H. erectus*, and that Stw 53 cannot. Because there is no clear indication that more than one species of *Homo* is represented in the Swartkrans sample, we consider all this material to belong to *H. erectus*. We considered '*Homo ergaster*' (and also '*Homo aff. erectus*' from *Wood, 1991*) to be synonyms of *H. erectus* for this study; Turkana Basin specimens that are attributed to *H. erectus* thus include KNM-ER 730, KNM-ER 820, KNM-ER 992, KNM-ER 1808, KNM-ER 3733, KNM-ER 3883, KNM-ER 42700, KNM-WT 15000. Olduvai specimens include OH 9, OH 12 and OH 28. Original fossil materials from Chemeron, Lake Turkana, Swartkrans, Trinil, and Dmanisi were examined first-hand by the authors, while the remainder were based on casts and published reports (*Weidenreich, 1943*; *Wood, 1991*; *Antón, 2003*; *Rightmire et al., 2006*; *Suwa et al., 2007*).

A large number of postcranial specimens have been collected from the Turkana Basin and appear consistent with the anatomical range otherwise found in *Homo*, and inconsistent with known samples of *Australopithecus* and *Paranthropus* from elsewhere. These include KNM-ER 1472, KNM-ER 1481, KNM-ER 3228, KNM-ER 737, and others. We may add other fossils from other sites lacking association with craniodental material, such as the partial BOU-VP 12/1 skeleton and even the Gona pelvis. These specimens attributable to *Homo* but not necessarily to a particular species did inform our understanding of variability within the genus, but for the most part these specimens do not inform our differential diagnosis of *H. naledi* relative to particular species. For example, the key element of femoral morphology of *H. naledi* in contrast to other species is the presence of two well-defined mediolaterally running pillars in the femoral neck; the isolated specimens of early *Homo* do not contradict this apparent autapomorphy. Likewise, no isolated specimens inform us about the humanlike aspects of foot morphology in *H. naledi*. In these cases, the lack of associations for this evidence actually is less important than the lack of specimens that replicate the distinctive features of the *H. naledi* morphology.

### Middle Pleistocene Homo

Specimens from the latest Lower Pleistocene and MP of Europe and Africa that cannot be attributed to *H. erectus* were included in our comparisons. These include fossils that have been attributed to *H. heidelbergensis*, *H. rhodesiensis*, 'archaic *H. sapiens*', or 'evolved *H. erectus*' by a variety of other authors. Specimens attributed to MP *Homo* include materials from Eliye Springs, Arago, Atapuerca Sima de los Huesos, Bodo, Broken Hill, Cave of Hearths, Ceprano, Dali, Elandsfontein, Jinniushan, Kapthurin, Mauer, Narmada, Ndutu, Petralona, Reilingen-Schwetzingen, Solo, Steinheim, Swanscombe. This grouping includes the following specimens: KNM-ES 11693, Arago 2, Arago 13, Arago 21, Atapuerca 1, Atapuerca 2, Atapuerca 4, Atapuerca 5, Atapuerca 6, Cave of Hearths, SAM-PQ-EH1, Kabwe, Mauer, Ndutu, Salé, Petralona, Reilingen-Schwetzingen, Steinheim.

### Homo floresiensis

Specimens from Liang Bua, Flores as described by *Brown et al., 2004*; *Morwood et al., 2005*, *Jungers et al., 2009a*, *Jungers et al., 2009b*, and *Falk et al., 2005* were included in this study.

## Scanning and virtual reconstruction methods

The calvariae (DH1-4) were scanned using a NextEngine laser surface scanner (NextEngine, Malibu, CA) at the following settings: Macro, 12 divisions with auto-rotation, HD 17k ppi. Depending on the

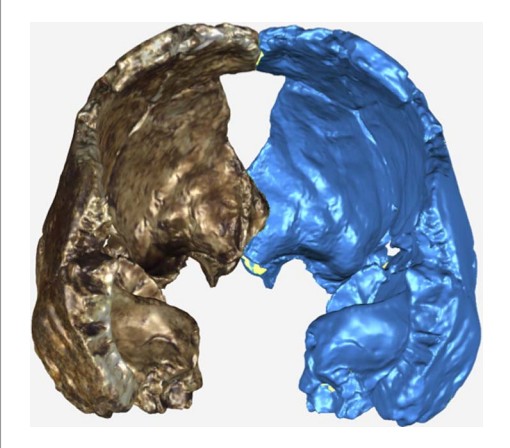

**Figure 15**. Posterior view of the virtual reconstruction of DH3. The resultant mirror image is displayed in blue. The antimeres were aligned by the frontal crest and sagittal suture using the Manual Registration function in GeoMagic Studio 14.0.

complexity of the surface relief, either two or three complete scanning cycles were completed per specimen, resulting in multiple 360° scans. Each individual scan was trimmed, aligned, and fused (volume merged) in the accompanying ScanStudio HD Pro software. For each specimen, the individual 360° scans were then aligned and merged in GeoMagic Studio 14.0 (Raindrop Geomagic, Research Triangle Park, NC), creating a final three-dimensional model of the specimen. Given the fragmented nature of the calvariae specimens, both the ectocranial and endocranial surfaces were captured in the scans.

DH3 consisted primarily of portions of the right calvaria. However, a small section of the frontal and the parietal crossed the mid–sagittal plane. For this reason, it was possible to mirror image the surface scan to approximate the left calvaria and obtain a more complete visualization of the complete calvaria (*Figure 15*). The virtual specimen of DH3 was mirrored in GeoMagic Studio, and manually registered (aligned) using common points along the frontal crest and sagittal suture. The registration procedure in GeoMagic Studio is an iterative process that refines the alignment of specimens to minimize spatial differences between corresponding surfaces. In this manner, the program is able to match the position overlapping surfaces, in addition to their angulation and curvature.

The same procedures were used to mirror image and create a virtual reconstruction of DH2 and the occipital portion of DH1 (*Figure 16*). The occipital and vault portions of DH1 were reconstructed based on the anatomical alignment of the sagittal suture, sagittal sulcus, parietal striae, and the continuation of the temporal lines across both the specimens.

## Virtual reconstruction of composite crania and estimation of cranial capacity

In order to virtually estimate the cranial capacity, composite crania were constructed from the surface scans and mirror imaged scans of the calvariae. Two separate composite crania were created; the relatively smaller-sized calvariae (DH3 and DH4) were combined into one composite, and the larger-sized calvariae (DH1 and DH2) composed the larger composite cranium.

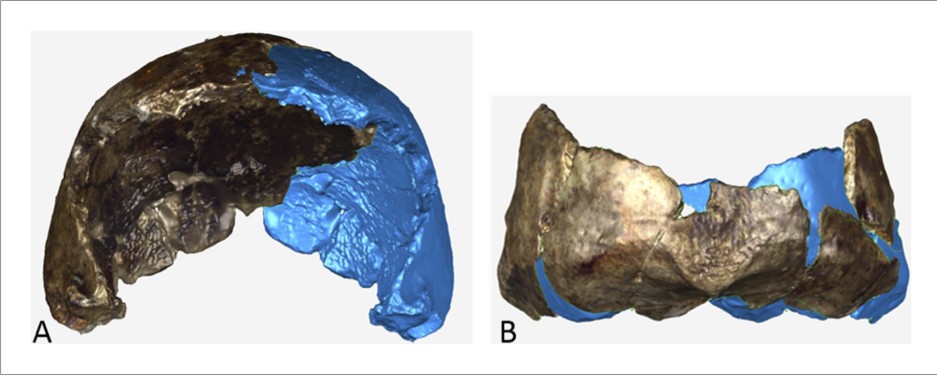

**Figure 16**. Virtual reconstruction of (**A**) DH2 and (**B**) occipital portion of DH1. The actual specimen displays its original coloration and the mirror imaged portion is illustrated in blue.

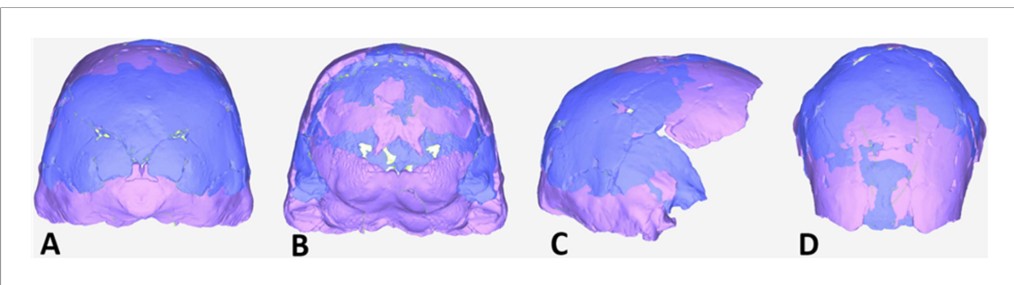

**Figure 17**. Postero-lateral view of the virtual reconstruction of a composite cranium from DH3 and DH4. (**A**) The surface scan of DH3 was mirror imaged and merged as described in Supplementary Note 8. (**B**) The scan of DH4 was aligned to the DH3 model. (**C**) DH4 was then mirror imaged to complete the occipital contour (**D**).

The smaller composite cranium, DH3 was mirrored in GeoMagic Studio 14.0, and merged with the original scan as outlined above. The surface scan of DH4 was uploaded and registered (aligned) to the DH3 model using overlapping temporal features (e.g., the external auditory meatus). No scaling was performed. DH4 was then mirror imaged to complete the occipital contour. The resultant model suggests a general concordance between the specimens in both size and shape with a close alignment of vault surfaces and anatomical features between specimens (*Figure 17*).

For the larger composite cranium, the surface model of DH2 and its mirror image was then uploaded, registered (aligned), and merged with the mirror-imaged model of DH1. No scaling was performed. The congruency between the specimens in the resultant model suggests that DH1 and DH2 are similar in both size and vault shape (*Figure 18*).

## Virtual reconstruction of cranial capacity

The composite model of DH3 and DH4 was used to estimate the cranial capacity for the smaller morphotype. In GeoMagic Studio 14.0, the endocranial surface of the composite was carefully selected from the ectocranial surface and copied as a new object. In order to obtain a volume calculation the model has to be a closed surface, meaning that all of the holes in the surface model had to be filled. Small holes in the model were filled using the 'Fill by Curvature' function. Larger holes were filled in by sections. For example, the cranial base was filled in using a number of transverse sections, so that in the absence of the cranial base the contour of the various cranial fossae and the petrous portions of the temporal could be preserved as best as possible. When appropriate (e.g., around angular portions of the petrous bone), small sections were filled using a flat hole filling function. The new surfaces created by the hole-filling mechanism were carefully monitored and repeated until an acceptable model that appeared to best approximate the missing portions was obtained. The result is a closed model approximation of the endocranium, of which a volume can be

**Figure 18**. Virtual reconstruction of a composite cranium from DH1 and DH2. The surface model of DH2 (blue), consisting of the original scan merged with the mirror image, was then uploaded and aligned with the mirror-imaged DH1 model (pink). Note the similarity in size and shape between DH1 and DH2 observed in the posterior (**A**) anterior (**B**) lateral (**C**) and superior (**D**) views.

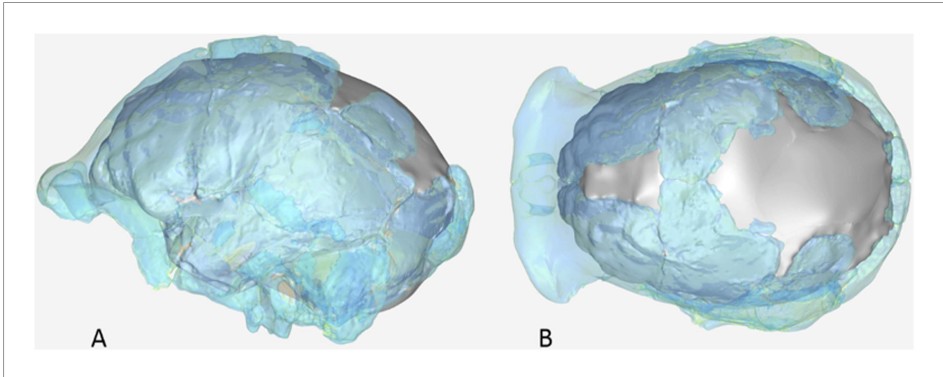

**Figure 19**. Virtual reconstruction of the endocranium of the composite cranium from DH3 and DH4. (**A**) Lateral view.
(**B**) Superior view. (**C**) Inferior view. In all views, anterior is to towards the left.

calculated by GeoMagic Studio (*Figure 19*, *Figure 20*). The volume of the smaller composite cranium
(DH3 and DH4) indicates a cranial capacity of approximately 465 cm³.

In order to determine whether significant errors were being introduced in the manner that the
cranial base was filled in the above procedures, the endocranial volume of DH3/DH4 was also virtually
calculated using the cranial base of Sts 19 as a model. A 3D model of Sts 19 was mirrored and aligned
to the DH3/DH4 model using the external auditory meatus and common points on the internal surface
of the petrous portion as a guide (*Figure 21*). The Sts 19 model was then scaled by 0.97 to obtain an
optimal fit between the two models.

After the Sts 19 model was merged with the DH3/DH4 model, the endocranial surface was
extracted and reconstructed as described above (*Figure 22*). The resultant endocranial volume using
the Sts 19 cranial base was 465.9 cm³. This value is in agreement with the first estimate and suggests
that using a model cranial base did not significantly alter the results.

The larger composite cranium, consisting of DH1 and DH2, lacks most of the frontal region. In
order to create a closed endocranial surface for a volume estimate, the frontal region from the smaller
composite cranium was scaled by 5%, and then registered (aligned) and merged to the model of the
larger composite cranium. As with the smaller composite cranium, the endocranial surface was then
selected and converted to a new object, and the remaining holes filled based on the curvature of the
surface. The volume of the closed endocranial model was calculated using GeoMagic Studio. The
cranial capacity (endocranial volume) of the larger composite model is approximately 560cc.

## Body mass estimation methods

Eight femoral fragments from the Dinaledi collection allow a direct measurement of the
subtrochanteric anteroposterior and mediolateral diameters (*Table 3*). We developed two regression

**Figure 20**. Virtual reconstruction of the endocranium of the composite cranium from DH3 and DH4 overlaid with the
ectocranial surfaces. (**A**) Lateral view. (**B**) Superior view.

**Figure 21**. Virtual reconstruction the DH3/DH4 cranial base using a model of Sts 19. (**A**) Right lateral view. (**B**) Left lateral view. (**C**) Posterior view. (**D**) Inferior view.

equations to estimate body mass from these diameters based on the masses of modern human samples. MCE measured body masses of a sample of 253 modern European individuals, 128 males and 125 females, collected from the Institute for Forensic Medicine in Zurich, Switzerland. Body masses were taken at time of forensic evaluation. This sample yields the following regression equation relating body mass to subtrochanteric diameter, where FSTpr refers to the product of the femoral subtrochanteric mediolateral and anteroposterior breadths:

$$\text{Body Mass} = 0.060 \times \text{FSTpr} + 13.856, \text{SEE} = 6.78, r = 0.50, p = <0.001.$$

We further examined a broader sample of 276 modern humans taken from a number of populations around the world, with data measured by TWH. The body masses of individuals were estimated from femur head diameter, using the average of results obtained from *Grine et al. (1995)* and *Ruff et al. (1997)*. The sample includes 115 females, 155 males, and 6 individuals of indeterminate sex.

$$\text{Body Mass} = 0.046 \times \text{FSTpr} + 24.614, \text{SEE} = 5.82, r = 0.82, p < 0.001.$$

## Stature estimation methods

We collected data from skeletal material representing two African population samples. We use only African populations in this comparison because the ratio of tibia length to femur length, and thereby the proportion of stature constituted by tibia length, varies between human populations both today and prehistorically. Although we do not know this proportion for *H. naledi*, we adopt the null hypothesis that they likely had tibia/femur proportions similar to other African population samples.

95 male and female Kulubnarti individuals from medieval Nubia are curated at the University of Colorado, Boulder. Data were collected by HMG, including estimates of living stature based on the Fully method (*Fully, 1956*; *Raxter et al., 2006*), and these were used to develop a regression equation relating tibia length to stature. The resulting equation is:

$$\text{Stature} = 0.295 \times \text{TML} + 48.589, \text{SEE} = 3.13, r = 0.90, p < 0.001.$$

We (HMG and TWH) collected measurements from 38 African males and 38 females curated within the Dart Collection of the University of the Witwatersrand. Specimens were randomly chosen with no preference for specific African ethnic groups. Cadaveric statures are documented for this collection, the regression equation relating tibia length to stature in this sample is:

$$\text{Stature} = 0.223 \times \text{TML} + 75.350, \text{SEE} = 6.50, r = 0.63, p < 0.001.$$

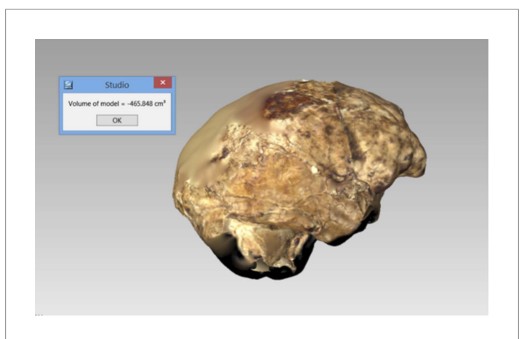

**Figure 22**. Virtual reconstruction the DH3/DH4 endocranial volume using a cranial base model of Sts 19. Right lateral view.

## Nomenclatural acts

The electronic edition of this article conforms to the requirements of the amended International Code of Zoological Nomenclature, and hence, the new name contained herein is available under that Code from the electronic edition of this article. This published work and the nomenclatural acts it contains have been registered in ZooBank, the online registration system for the ICZN. The ZooBank LSIDs (Life Science Identifiers) can be resolved and the associated information viewed through any standard web browser by appending the LSID to the prefix 'http://zoobank.org/'. The LSID for this publication is: urn:lsid:zoobank.org:pub:00D1E81A-6E08-4A01-BD98-79A2CEAE2411. The electronic edition of this work was published in a journal with an ISSN (2050-084X) and has been archived and is available from the following digital repositories: PubMed Central and LOCKSS.

## Access to material

All Dinaledi fossil material is available for study by researchers upon application to the Evolutionary Studies Institute at the University of the Witwatersrand where the material is curated (contact Bernhard Zipfel [Bernhard.Zipfel@wits.ac.za]). Three-dimensional surface renderings and other digital data are available from the MorphoSource digital repository (http://morphosource.org).

## Acknowledgements

The authors would like to thank the many funding agencies that supported various aspects of this work. In particular the authors would like to thank the National Geographic Society, the South African National Research Foundation and the Gauteng Provincial Government for particularly significant funding of the discovery, recovery and analysis of this material. Other funding agencies include and the Palaeontological Scientific Trust, the Texas A&M College of Liberal Arts Seed Grant Program, the Lyda Hill Foundation and the Wisconsin Alumni Research Foundation. We wish to thank the Jacobs Family for access to the site and the South African Heritage Resource Agency and Cradle of Humankind UNESCO World Heritage Site Management Authority for issuing the various permits required for this work, including the excavation permit (PermitID: 952). We would also like to thank the University of the Witwatersrand and the Evolutionary Studies Institute as well as the South African National Centre of Excellence in PalaeoSciences for curating the material and hosting the authors while studying the material.

## Additional information

### Funding

| Funder | Author |
| --- | --- |
| National Geographic Society | Lee R Berger |
| The National Research Foundation of South Africa | Lee R Berger |
| The Palaeontological Scientific Trust | Lee R Berger |
| Lyda Hill Foundation | Lee R Berger |
| Wisconsin Alumni Research Foundation (WARF) | John Hawks |
| Texas A and M University | Darryl J de Ruiter |

The funders had no role in study design, data collection and interpretation, or the decision to submit the work for publication.

### Author contributions

LRB, JH, DJR, SEC, Conception and design, Acquisition of data, Analysis and interpretation of data, Drafting or revising the article, Contributed unpublished essential data or reagents; PS, Conception and design, Acquisition of data, Analysis and interpretation of data, Contributed unpublished essential data or reagents; LKD, TLK, HMG, SAW, JMDS, MMS, CMM, NC, TWH, WH-S, MB, BB, DB, JB, ZDC, KAC, ASD, MD, MCE, EMF, DG-M, DJG, AG, JDI, MFL, DM, MRM, CMO, DR, LS, JES, ZT, MWT, CVS, CSW, Acquisition of data, Analysis and interpretation of data, Contributed unpublished

essential data or reagents; RRA, MD, Analysis and interpretation of data, Contributed unpublished essential data or reagents; AK, SN, EWN, PW, BZ, Acquisition of data, Analysis and interpretation of data

---

## Additional files

### Supplementary files
• Supplementary file 1. Holotype and paratype specimens and referred materials.

• Supplementary file 2. Traits of *H. naledi* and comparative species.

---

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
