## [Decision Letter]

Thank you for submitting your work entitled “A new species of the genus *Homo* from the Dinaledi Chamber, South Africa” for peer review at *eLife.* Your submission has been favorably evaluated by Ian Baldwin (Senior editor), two guest Reviewing editors (Johannes Krause and Nicholas Conard), and two peer reviewers. One of the two peer reviewers, Chris Stringer, has agreed to share his identity, and Johannes Krause has drafted this decision to help you prepare a revised submission.

The authors describe a large collection of recently discovered hominin fossils from the Dinaledi Chamber in the Rising Star cave system in South Africa. Based on their initial assessment they argue that the fossil remains derive from a single homogenous hominin group and present a new taxon that they call *Homo naledi*.

Given the importance and sheer number of hominin remains, the fossils from the Dinaledi cave should be described and published imminently. This will allow direct assess to the material to other researchers where appropriate. Besides a general agreement among the reviewers for publication, they ask for several essential revisions.

1) The reviewers are surprised to not see an in-depth comparison of *H. naledi* to *H. floresiensis*, especially where combinations of small teeth and small brains are concerned. It should be easy, e.g., to add the published *H. floresiensis* measurements to Figure 7. The authors allude to material attributed to *‘Homo gautengensis’* and perhaps a short discussion or reiteration of their views about the validity of that species is needed.

2) The reviewers would ask the authors to remove superfluous text and concentrate on the main finding and description. This includes the first and last paragraphs. (“Our view of human evolution ... ancient hominins.”) (“Decades of work remain to expand the window into our origins...”) Both do not add any essential information to the manuscript.

3) The statement that “At this time, this evidence does not permit us to resolve the relationships between *H. naledi* and these other species” is rather confusing. Certainly more remains or a direct date would not change that situation; instead in depth morphometric or cladistics analysis are needed to conclude whether the Dinaledi remains indeed represent a new hominin taxon.

4) The authors should add a summary table of traits, perhaps adapted from the one they previously used for cladistic assessment.

In addition, please note that Chris Stringer has provided an annotated PDF with minor comments for your consideration.

---

## [Author Response]

*1) The reviewers are surprised to not see an in-depth comparison of* H. naledi *to* H. floresiensis*, especially where combinations of small teeth and small brains are concerned. It should be easy, e.g., to add the published* H. floresiensis *measurements to*
Figure 7*. The authors allude to material attributed to* ‘Homo gautengensis’ *and perhaps a short discussion or reiteration of their views about the validity of that species is needed*.

We have added *H. floresiensis* to the differential diagnosis, including relevant aspects of cranial, dental, hand, lower limb and foot anatomy. We have also added *H. floresiensis* to both relevant figures (Figure 5 and Figure 7). We have also made a short comment on the validity of *H. gautengensis* in the Materials and methods.

*2) The reviewers would ask the authors to remove superfluous text and concentrate on the main finding and description. This includes the first and last paragraphs. (“Our view of human evolution ... ancient hominins.”) (“Decades of work remain to expand the window into our origins...”) Both do not add any essential information to the manuscript*.

We have removed both initial and final paragraphs in their entirety as suggested, and have slightly altered the remaining (second and penultimate) paragraphs to introduce and conclude the paper.

*3) The statement that “At this time, this evidence does not permit us to resolve the relationships between* H. naledi *and these other species” is rather confusing. Certainly more remains or a direct date would not change that situation; instead in depth morphometric or cladistics analysis are needed to conclude whether the Dinaledi remains indeed represent a new hominin taxon*.

We have altered this aspect of the Discussion in order to clarify what we can say about the phylogenetic placement of *H. naledi*. This includes a timely reference to a new paper on the phylogeny of hominins, which allows us to contextualize the difficulty placing the species considering the mosaic of similarities and differences from known hominins.

*4) The authors should add a summary table of traits, perhaps adapted from the one they previously used for cladistic assessment*.

This summary table is now added as Table 2; the remaining tables have been renumbered accordingly.

*In addition, please note that Chris Stringer has provided an annotated PDF with minor comments for your consideration*.

We have taken on board all of Chris Stringer’s helpful comments and editorial suggestions, and included these additions and corrections in the manuscript. We did not make changes to the following two suggestions/questions for the following reasons:

“*Any comment possible on frontal sinus morphologies*?”

None are adequately preserved for us to be confident in making descriptive comments at this time.

“*Anything to say on middle or inner ear morphology*?”

We are not in a position to comment on inner ear morphology as this is a more detailed and long-term study for which significant comparative material must be assembled.